# DOMAIN GENERALIZATION DEEP GRAPH TRANSFOR-MATION

## ABSTRACT

Graph transformation that predicts graph transition from one mode to another is an important and common problem. Despite much recent progress in developing advanced graph transformation techniques, the fundamental assumption typically required in machine-learning models that the testing and training data preserve the same distribution does not always hold. As a result, domain generalization graph transformation that predicts graphs not available in the training data is under-explored, with multiple key challenges to be addressed including (1) the extreme space complexity when training on all input-output mode combinations, (2) difference of graph topologies between the input and the output modes, and (3) how to generalize the model to target domains that are not in the training data. To fill the gap, we propose a multi-input, multi-output, hypernetwork-based graph neural network (MultiHyperGNN) that employs a encoder and a decoder to encode both input and output modes and semi-supervised link prediction to enhance the graph transformation task. Instead of training on all mode combinations, MultiHyperGNN preserves a constant space and polynomial computational complexity with the encoder and the decoder produced by two novel hypernetworks. Comprehensive experiments show that MultiHyperGNN has a superior performance than competing models in both prediction and domain generalization tasks.

## 1 INTRODUCTION

Graph is a ubiquitous data structure characterized by node attributes and the graph topology that describe objects and their relationships. Many tasks on graphs ask for predicting a graph (i.e., graph topology or node attributes) from another one. Applications of such graph transformation include traffic forecasting between two time stamps based on traffic flow (Li et al., 2018; Yu et al., 2018), fraud detection between transactional periods (Van Belle et al., 2022), and chemical reaction prediction according to molecular structures (Guo et al., 2019).

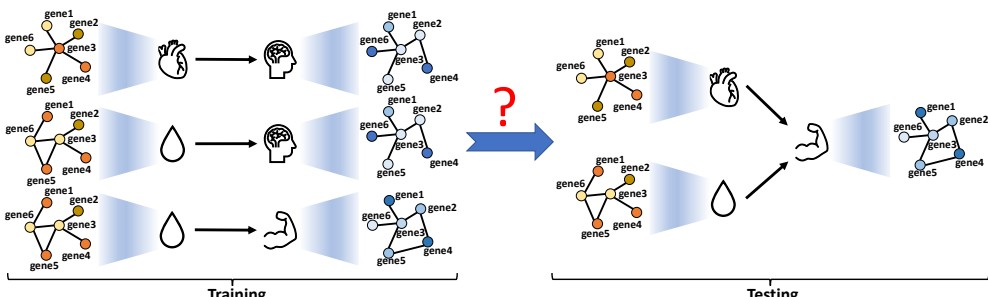

Figure 1: Graph transformation predicts gene expression of the output tissue from that of the input tissue. The yellow and the blue networks are gene-gene networks in the input and the output tissues, respectively. Node colors indicate various expression levels. We train the model on input-target tissue pairs, whereas we generalize the model to unseen mode pairs or even unseen modes during testing.

Despite of a wide spectrum of applications, graph transformation still faces major issues such as insufficient samples of graph pairs for training the model. For instance, as shown in Figure 1, if the model is trained to predict gene-gene network on specific tissue pairs (e.g., from heart and blood

to brain, from blood to muscle), but in testing process, one may want to generalize the model to unseen tissue pairs (e.g., from heart to muscle) or even to tissues unavailable in the training data. If so, the performance of the graph transformation model may deteriorate due to domain distribution gaps (Quinonero-Candela et al., 2008). Therefore, it is imperative and crucial to improve the generalization ability of graph transformation models to generalize the learned graph transformation to other (unseen) graph transformations, namely domain generalization graph transformation.

Domain generalization graph transformation, nevertheless, is still under-explored by the machine-learning community due to the following challenges: **(1) High complexity in the training process**. To learn the distribution of graph (or mode) pairs in training data, we need to learn the model by traversing on all combinations of input modes to predict all combinations of output modes. In this case, the training complexity would be exponential if we train a single model for all possible input-output mode combinations; **(2) Graph transformation between topologically different modes**. The existing works regarding graph transformation predict node attributes conditioning on either the same topology or the same set of nodes of input and output modes (Battaglia et al., 2016; Yu et al., 2018; Guo et al., 2019). Performing graph transformation across modes with varying topologies, including different edges and even varying graph sizes, is a difficult task. Main challenges include how to learn the mapping between distinct topologies and how to incorporate the topology of each mode to enhance the prediction task; **(3) Learning graph transformation involving unseen domains and lack of training data.** Graph transformation usually requires both the source and target domains to be visible and have adequate training data to train a sophisticated model. However, during the prediction phase, we may be asked to predict a graph in an unseen target domain. Learning such transformation mapping without any training data is an exceedingly challenging task.

To fill the gap, we propose a novel framework for domain generalization graph transformation via a multi-input, multi-output hypernetwork-based graph neural networks (MultiHyperGNN). Our contributions are summarized as follows:

- We propose a novel multi-input, multi-output framework of graph transformation for predicting node attributes across multiple input and output modes. We introduced a novel framework leveraging a multi-input, multi-output training strategy, significantly reducing the space complexity regarding trainable parameters from exponential to constant during training.

- We develop an encoder and a decoder for graph transformation between topologically different input and output modes, respectively. Additionally, our model conducts semi-supervised link prediction to complete the output graph, facilitating generalization to all nodes in the output mode.

- We design two novel hypernetworks that produce the encoder and the decoder for domain generalization. Mode-specific meta information serves as the input to guide the hypernetwork to produce the corresponding encoder or decoder, and generalize to unseen target domains.

- We conduct extensive experiments to demonstrate the effectiveness of MultiHyperGNN on three real-world datasets. The results show that MultiHyperGNN is superior than competing models.

## 2 RELATED WORKS

**Graph transformation**. Graph transformation maps graph from one mode to another. Existing works either predict node attributes given fixed graph topology, such as Li et al. Li et al. (2018) in traffic forecasting under fixed traffic network and Battaglia et al. Battaglia et al. (2016) in predicting velocities of objects on subsequent time steps, or predict graph topology, such as Guo et al. Guo et al. (2022) learning by the global and the local translation with graph convolution and deconvolution layers. Others instead predict node attributes and graph topology simultaneously. Guo et al. Guo et al. (2019) solved node-edge joint translation with a multi-block network. Lin et al. (Lin et al., 2020) applied graph attention to the co-evolution of node and edge states. When predicting node attributes, nevertheless, the assumption of fixed graph topology in both input and output modes may not always hold. Graph transformation that can handle topologies of both modes is needed.

**Domain generalization**. Machine learning models typically assume identical distribution between training and testing data. However, model generalization to unseen data is crucial in fields like semantic segmentation (Gong et al., 2019; Dou et al., 2019), fault diagnosis (Zheng et al., 2020), and natural language processing (Wang et al., 2020; Garg et al., 2021), among others (Du et al., 2021; Qian et al., 2021). For instance, Du et al.Du et al. (2021) tackled temporal covariate shift in time series using an RNN-based model, while Qian et al.(Qian et al., 2021) addressed sensor-based

activity recognition by learning domain-invariant modules. Similarly, Gong et al.(Gong et al., 2019) focused on domain generalization in image translation, Wang et al.(Wang et al., 2020) employed meta-learning for zero-shot domain generalization in semantic parsing, and Chen et al. (Chen et al., 2022) explored latent domain structure identification without domain labels, emphasizing the need for expressive representation learning.

**Hypernetworks**. A hypernetwork is a neural network generating weights for another network (Ha et al., 2017), with applications spanning image classification (Sun et al., 2017; Sendera et al., 2023), editing (Alaluf et al., 2022), robotic control (Huang et al., 2021; Rezaei-Shoshtari et al., 2023), and language models (Volk et al., 2022; Zhang et al., 2022). Notably, hypernetworks have been leveraged for domain generalization. Qu et al.(Qu et al., 2022) enabled expert weight generation sharing meta-knowledge, albeit with the training space complexity linear to the number of classifiers. Sendera et al.(Sendera et al., 2023) introduced HyperShot for few-shot learning by feeding kernel-based support representations to hypernetworks. Bai et al. Bai et al. (2022) employed hypernetworks for graph classifier generation using only timestamps for temporal domain generalization. Despite their widespread use in domain generalization, hypernetwork research in generating GNNs is sparse. Therefore, we introduce two novel hypernetworks to steer domain generalization in graph transformation tasks.

## 3 PROBLEM FORMULATION

Suppose we have $N$ modes of graphs composed of $p$ nodes: $\mathcal{G} = \{\mathcal{G}^{(1)}, \mathcal{G}^{(2)}, ..., \mathcal{G}^{(N)}\}$, where each mode contains graphs with the same topology. Specifically, suppose there are $n$ independent samples in the dataset, and for each sample $i$ in the mode $j$, denote $G_i^{(j)} = \{A^{(j)}, X_i^{(j)}\} \in \mathcal{G}^{(j)}$, where $A^{(j)} \in \mathbb{R}^{p_j \times p_j}$ is the adjacency matrix of size $p_j \leq p$ and $X_i^{(j)} \in \mathbb{R}^{p_j \times d}$ is the node attributes with $d$ features. Note that the graph of $\mathcal{G}^{(j)}$ may not contain all $p$ nodes in mode $j$, all other nodes are disjointed with each other and with $p_j$ nodes in $\mathcal{G}^{(j)}$. We further assume each mode $j$ can be characterized by its meta information $m^{(j)}$. For instance, the mode $\mathcal{G}^{(j)}$ can be a specific human tissue $j$ that has the gene-gene expression network $G_i^{(j)}$ for a patient $i$. There are $p$ human genes expressed in various human tissues but $G_i^{(j)}$ only contains $p_j$ of them. A detailed notation table is in Appendix A.

Next, we formally formulate the task **domain generalization graph transformation** as below:

**Definition 1** (Domain generalization graph transformation). Let $\mathcal{S} = \{\mathcal{X} \times \mathcal{Y} : \mathcal{X} \in \mathcal{P}(\mathcal{G}), \mathcal{Y} \in \mathcal{P}(\mathcal{G} - \mathcal{X})\}$ be the source domain where we train the graph-transformation model $f : \mathcal{X} \to \mathcal{Y}$, which predicts node attributes in $\mathcal{Y}$ from node attributes in $\mathcal{X}$. $\mathcal{P}(\cdot)$ is the power set excluding the empty set. *Domain generalization graph transformation* learns the $f$ so that the prediction error on $f : \mathcal{X}^{\mathcal{T}} \to \mathcal{Y}^{\mathcal{T}}$ is minimized, where $\mathcal{X}^{\mathcal{T}} \times \mathcal{Y}^{\mathcal{T}}$ is the target domain s.t. $\mathcal{X}^{\mathcal{T}} \times \mathcal{Y}^{\mathcal{T}} \notin \mathcal{S}$.

Domain generalization graph transformation is exceptionally difficult due to the following challenges:

**Challenge 1: High complexity in training process.** For training the graph transformation $f : \mathcal{X} \to \mathcal{Y}$, where $\mathcal{X} \times \mathcal{Y} \in \mathcal{S}$, conventionally we need to train $O(3^N)$ models to handle all possible mode combinations in $\mathcal{S}$, which is rather computationally intensive.

**Challenge 2: Topological difference between input and output domains.** When the input mode and the output mode have different topologies, how to utilize topologies of both modes to jointly contribute to graph transformation remains to be explored. An intuitive way is to employ two graph encoders to respectively encode the graph topology of both modes, but how to form the graph-transformation model on $\mathcal{S}$ with only two such encoders is still challenging.

**Challenge 3: Generalization to unseen domains.** Even if it is possible that we train the model on all combinations of modes in $\mathcal{S}$, how to learn the graph transformation that can efficiently predict the graph in an unseen target domain is still challenging.

## 4 DOMAIN GENERALIZATION DEEP GRAPH TRANSFORMATION

### 4.1 OVERVIEW OF MULTIHYPERGNN

For the first challenge, instead of including exponentially many modes by separately training on all their combinations, we collectively train all modes together to avoid the duplication of modes and reduce the complexity (Figure 2). To address the heterogeneity of nodes and graph topology, we propose an encoder-decoder framework in Section 4.2 (Figure 2 (A)). Moreover, each input

mode requires an encoder while each output mode needs a decoder, which can be any type of GNNs such as Graph Convolutional Network (GCN), Graph Isomorphism Network (GIN) and Graph Attention Network (GAT). To learn the encoder and the decoder for unseen modes, we propose to train two hypernetworks that respectively generate any encoder or decoder given mode-specific meta information in Section 4.3 (Figure 2 (B)). Furthermore, we provide a theoretical assurance that an ample amount of meta-information will result in improved generalization accuracy when extrapolating to unexplored domains.

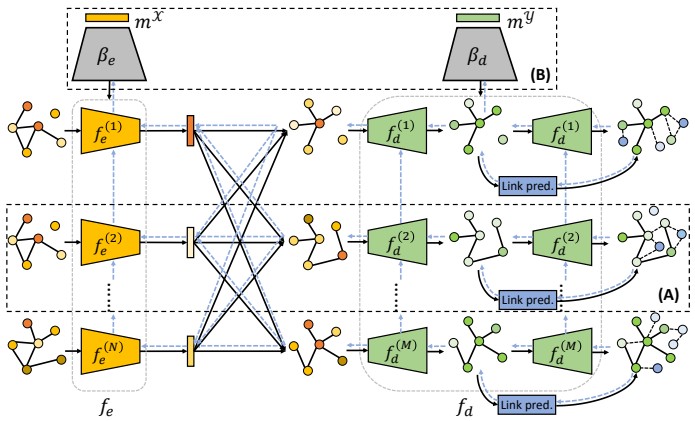

Figure 2: Overview of MultiHyperGNN for the graph transformation from $N$ input to $M$ output modes. $\{f_e^{(j)}\}_{j \in \mathcal{X}}$ are encoders generated by the encoder hypernetwork $\beta_e$. $\{f_d^{(k)}\}_{k \in \mathcal{Y}}$ are decoders generated by the decoder hypernetwork $\beta_d$. $m^{\mathcal{X}}$ and $m^{\mathcal{Y}}$ are mode-specific meta information. "Link pred." is semi-supervised link prediction. Blue dotted line is the flow of back-propagation.

Let $j \in \mathcal{X}$ be the $j$-th mode in $\mathcal{X}$ and $k \in \mathcal{Y}$ be the $k$-th mode in $\mathcal{Y}$, where $\mathcal{X} \in \mathcal{P}(\mathcal{G})$ and $\mathcal{Y} \in \mathcal{P}(\mathcal{G} - \mathcal{X})$ as in Def. 1. As in Figure 2 (A), to predict $X_i^{(k)}$, we use encoders (i.e., $f_e = \{f_e^{(j)}\}_{j \in \mathcal{X}}$) and decoders (i.e., $f_d = \{f_d^{(k)}\}_{k \in \mathcal{Y}}$) to encode the topology of both input and output modes:

$$\hat{X}_i^{(k)} = f_d^{(k)}(A^{(k)}, \sigma_1(\{f_e^{(j)}(G_i^{(j)}; \beta_e^{(j)})\}_{j \in \mathcal{X}}); \beta_d^{(k)})), \quad k \in \mathcal{Y} \tag{1}$$

Namely, to predict node attributes of any $k \in \mathcal{Y}$ from modes in $\mathcal{X}$, we first encode topologies of modes in $\mathcal{X}$ via $f_e^{(j)}(G_i^{(j)}; \beta_e^{(j)})$, $j \in \mathcal{X}$. Then we aggregate embeddings of all these modes via the pooling function $\sigma_1(\cdot)$ and feed it with the graph topology $A^{(k)}$ into $f_d^{(k)}$ to predict $X_i^{(k)}$, where $k \in \mathcal{Y}$. To reduce the heavy complexity of the training process due to the exponential number of choices of $\mathcal{X} \times \mathcal{Y} \in \mathcal{S}$ (Def. 1) and generalize the mode to unseen domains, instead of training separately for each $\mathcal{X} \times \mathcal{Y}$, as shown in Figure 2 (B), we borrow two hypernetworks (i.e., $\beta_e$ and $\beta_d$) to produce all encoders and decoders with the corresponding mode-specific meta information:

$$\beta_e^{(j)} = \beta_e(m^{(j)}; \gamma_e), \quad j \in \mathcal{X}; \quad \beta_d^{(k)} = \beta_d(m^{(k)}; \gamma_d), \quad k \in \mathcal{Y}, \tag{2}$$

where $\gamma_e$ and $\gamma_d$ parameterize $\beta_e$ and $\beta_d$, respectively, and are learned during training process. Therefore, Eq. 1 is re-parameterized by $\gamma_{\mathcal{X} \to \mathcal{Y}} = \{\gamma_e, \gamma_d\}$:

$$\begin{aligned} \hat{X}_i^{(k)} &= f_d^{(k)}(A^{(k)}, \sigma_1(\{f_e^{(j)}(G_i^{(j)}; \beta_e(m^{(j)}; \gamma_e))\}_{j \in \mathcal{X}}); \beta_d(m^{(k)}; \gamma_d)), \quad k \in \mathcal{Y} \\ &= f_{\gamma_{\mathcal{X} \to \mathcal{Y}}}(A^{(k)}, \{G_i^{(j)}, m^{(j)}\}_{j \in \mathcal{X}}; \gamma_{\mathcal{X} \to \mathcal{Y}}), \quad k \in \mathcal{Y}, \end{aligned} \tag{3}$$

where $f_{\gamma_{\mathcal{X} \to \mathcal{Y}}} = \{f_d^{(k)} * \{f_e^{(j)}\}_{j \in \mathcal{X}}\}_{k \in \mathcal{Y}} : \mathbb{R}^{N \times p} \to \mathbb{R}^{M \times p}$ formularizes the graph transformation that predicts node attributes of $M$ modes in $\mathcal{Y}$ from $N$ modes in $\mathcal{X}$.

As long as $\hat{X}_i^{(k)}$ is predicted via Eq. 3, we mathematically formulate the first term of the learning objective of MultiHyperGNN as follows:

$$\mathcal{L}_1(\gamma_e, \gamma_d) = \sum_{i=1}^{n} \ell(\{\hat{X}_i^{(k)}\}_{k \in \mathcal{Y}}, \{X_i^{(k)}\}_{k \in \mathcal{Y}}) \tag{4}$$

where $\ell(\cdot)$ measures the prediction error of $f_{\gamma_{\mathcal{X} \to \mathcal{Y}}}$ of each sample, such as mean squared error (MSE), mean absolute error (MAE), etc. $n$ is the total number of samples in training data.

Since the size of the source domain $\mathcal{S}$ is $O(3^N)$ (i.e., each of $N$ modes can serve as source mode, target mode or neither), leading to an exponential space complexity of $O(3^N)$ with the space of trainable parameters as $\{\mathcal{P}(\{\beta_e^{(j)}\}_{j \in \mathcal{X}}) \times \mathcal{P}(\{\beta_d^{(k)}\}_{k \in \mathcal{Y}}) : \mathcal{X} \in \mathcal{P}(\mathcal{G}), \mathcal{Y} \in \mathcal{P}(\mathcal{G} - \mathcal{X})\}$. MultihyperGNN reduces the space complexity to $O(1)$ even though it can process arbitrary combinations of input and output modes during training.

## 4.2 Graph transformation on topologically different domains

Traditional graph-transformation models encounter significant challenges when attempting to handle modes with different graph topologies (i.e., $A^{(j)} \neq A^{(k)}, p_j \neq p_k$). To address this issue, as shown in Figure 2 (A), we propose GNN-based encoder $f_e^{(j)}$ and decoder $f_d^{(k)}$ that encode the graph of modes $j \in \mathcal{X}$ and $k \in \mathcal{Y}$, perform semi-supervised link prediction to complete the topology of the output mode $\mathcal{G}^{(k)}$ and enable the model to predict all $p$ nodes. Let $\mathcal{V}^{(j)}$ and $\mathcal{V}^{(k)}$ be sets of nodes contained in the graph of modes $j$ and $k$, respectively, and $|\mathcal{V}^{(j)}| = p_j, |\mathcal{V}^{(k)}| = p_k$. Since $p_j \neq p_k$, to match the input dimension of $f_e^{(j)}$ and $f_d^{(k)}$, we expand $\mathcal{G}^{(j)}$ and $\mathcal{G}^{(k)}$ by the union of their nodes and obtain $\tilde{\mathcal{G}}^{(j)}$ and $\tilde{\mathcal{G}}^{(k)}$ with node sets: $\tilde{\mathcal{V}}^{(j)} = \tilde{\mathcal{V}}^{(k)} = \mathcal{V}^{(j)} \bigcup \mathcal{V}^{(k)}$, and $|\tilde{\mathcal{V}}^{(j)}| = \tilde{p}_j = |\tilde{\mathcal{V}}^{(k)}| = \tilde{p}_k \leq p$. Those newly added nodes are self-connected and are disjointed with other nodes.

**Encoder**. For the $i$-th sample, the encoder $f_e^{(j)}$ encodes the topology and node attributes of the mode $j$ into the latent embedding $\mathbf{z}_i^{(j)} \in \mathbb{R}^l$, where $l$ is the hidden dimension:

$$\mathbf{z}_i^{(j)} = f_e^{(j)}(\tilde{G}_i^{(j)}; \beta_e^{(j)}) = \text{GNN}(\tilde{G}_i^{(j)}; \beta_e^{(j)}). \tag{5}$$

Based on Eq. 2, the encoder $f_e^{(j)}$ is generated by the hypernetwork $\beta_e$ guided by the mode-specific meta information $m^{(j)}$. Therefore, Eq. 5 becomes $\mathbf{z}_i^{(j)} = \text{GNN}(\tilde{G}_i^{(j)}; \beta_e(m^{(j)}; \gamma_e))$, where $\gamma_e$ is mode-invariant and parameterizes all encoders $\{f_e^{(j)}\}_{j \in \mathcal{X}}$.

**Decoder**. Once $\{\mathbf{z}_i^{(j)}\}_{j \in \mathcal{X}}$ is obtained for all modes in $\mathcal{X}$, we apply the decoder $f_d^{(k)}$ to decode $\{\mathbf{z}_i^{(j)}\}_{j \in \mathcal{X}}$ and encode the topology $\tilde{A}^{(k)}$ of the output mode $k \in \mathcal{Y}$ to predict node attributes of $\tilde{\mathcal{V}}^{(k)}$:

$$\hat{\tilde{X}}_i^{(k)} = f_d^{(k)}(\tilde{A}^{(k)}, \sigma_1(\{\mathbf{z}_i^{(j)}\}_{j \in \mathcal{X}}); \beta_d^{(k)}) = \text{MLP}(\text{GNN}(\tilde{A}^{(k)}, \sigma_1(\{\mathbf{z}_i^{(j)}\}_{j \in \mathcal{X}}); \beta_{d,\text{GNN}}^{(k)}); \beta_{d,\text{MLP}}^{(k)}), \tag{6}$$

where the Multilayer Perceptron (MLP) serves as the prediction layer and $\beta_d^{(k)} = \{\beta_{d,\text{GNN}}^{(k)}, \beta_{d,\text{MLP}}^{(k)}\}$, generated by the hypernetwork $\beta_d$ with mode-specific meta information $m^{(k)}$. Then Eq. 6 becomes:

$$\hat{\tilde{X}}_i^{(k)} = \text{MLP}(\text{GNN}(\tilde{A}^{(k)}, \sigma_1(\{\mathbf{z}_i^{(j)}\}_{j \in \mathcal{X}}); \beta_d(m^{(k)}; \gamma_d)); \beta_d(m^{(k)}; \gamma_d)), \tag{7}$$

where $\gamma_d$ is model-invariant and parameterizes all target decoders $\{f_d^{(k)}\}_{k \in \mathcal{Y}}$. We further define $\mathcal{V}$ as the set of all nodes contained in $\mathcal{G}$ so that $|\mathcal{V}| = p$. Since $p \geq \tilde{p}_j = \tilde{p}_k$, now we have only predicted node attributes of $\tilde{\mathcal{V}}^{(k)}$, and the attributes of the remaining nodes $\mathcal{V} \setminus \tilde{\mathcal{V}}^{(k)}$ still need to be predicted. **Semi-supervised link prediction**. We adopt the semi-supervised link prediction to complete the topology $\hat{A}^{(k)}$ of the mode $k$ using graph auto-encoder (Kipf & Welling, 2016) supervised by $\tilde{A}^{(k)}$:

$$\mathbf{h}_i^{(k)} = \text{GNN}(\tilde{A}^{(k)}, \hat{\tilde{X}}_i^{(k)}; \phi), \quad \hat{A}^{(k)} = \text{Sigmoid}(\mathbf{h}_i^{(k)} \cdot (\mathbf{h}_i^{(k)})^T) \tag{8}$$

Then we compute the Binary Cross Entropy (BCE) between $\tilde{A}^{(k)}$ and $\hat{A}^{(k)}$ as the second term of the learning objective:

$$\mathcal{L}_2(\phi) = \sum_{k \in \mathcal{Y}} \text{BCE}(\tilde{A}^{(k)}, \hat{A}^{(k)}) = -\sum_{k \in \mathcal{Y}} \sum_{s=1}^{p_k} \sum_{t=1}^{p_k} \{\tilde{A}_{st}^{(k)} \log \hat{A}_{st}^{(k)} + (1 - \tilde{A}_{st}^{(k)}) \log(1 - \hat{A}_{st}^{(k)})\} \tag{9}$$

Once Eq. 8 is trained and $\hat{\phi}$ is learned, we perform link prediction and update $\tilde{A}^{(k)}$ as follows:

$$\bar{\mathbf{h}}_i^{(k)} = \text{GNN}(\bar{A}^{(k)}, \bar{X}_i^{(j \to k)}; \hat{\phi}), \quad \tilde{A}^{(k)} \leftarrow \text{Sigmoid}(\bar{\mathbf{h}}^{(k)} \cdot (\bar{\mathbf{h}}^{(k)})^T), \tag{10}$$

where $\bar{A}^{(k)}$ is the diagonal block matrix with $\tilde{A}^{(k)}$ and the identity matrix $I \in \mathbb{R}^{(p - \tilde{p}_k) \times (p - \tilde{p}_k)}$ as diagonal blocks. Since the node attributes of $\tilde{\mathcal{V}}^{(k)}$ has been predicted in Eq. 7, we only need to impute the attributes of $\mathcal{V} \setminus \tilde{\mathcal{V}}^{(k)}$ with the corresponding attributes of modes in $\mathcal{X}$ as the input of GNN($\cdot$). Therefore, $\bar{X}_i^{(j \to k)} = [\hat{\tilde{X}}_i^{(k)}, \sigma_2(\{X_i^{(j)}[\tilde{p}_k :]\}_{j \in \mathcal{X}})]$ is the concatenation of previously predicted

attributes $\hat{\tilde{X}}^{(k)}$ (Eq. 7) and the aggregated attributes of $\mathcal{V} \setminus \tilde{\mathcal{V}}^{(k)}$ in modes $j \in \mathcal{X}$ via the pooling function $\sigma_2(\cdot)$. $\sigma_2(\cdot)$ is the mean pooling function across modes in $\mathcal{X}$ in implementation.

Once $\tilde{A}^{(k)}$ is updated by Eq. 10, we apply the decoder again in Eq. 7 to predict attributes of $\mathcal{V} \setminus \tilde{\mathcal{V}}^{(k)}$:

$$\hat{\hat{X}}_i^{(k)} = \text{MLP}(\text{GNN}(\tilde{A}^{(k)}, \bar{X}_i^{(j \to k)}; \beta_d(m^{(k)}; \gamma_d)); \beta_d(m^{(k)}; \gamma_d))[\tilde{p}_k :], \tag{11}$$

Finally, the predicted node attributes in $k \in \mathcal{Y}$ are $\hat{X}_i^{(k)} = [\hat{\hat{X}}_i^{(k)}, \hat{\tilde{X}}_i^{(k)}]$.

### 4.3 Domain generalization via hypernetworks

In this section, we propose the encoder hypernetwork ($\beta_e$ in Figure 2 (B)), the decoder hypernetwork ($\beta_d$ in Figure 2 (B)), and the algorithm to learn them. The characteristics and similarity among input and output modes is captured by meta information $m^{\mathcal{X}} = \{m^{(j)}\}_{j \in \mathcal{X}}$ and $m^{\mathcal{Y}} = \{m^{(k)}\}_{j \in \mathcal{Y}}$, respectively, which guide the encoder and the decoder hypernetwork ($\beta_e$ and $\beta_d$, respectively) to produce mode-specific encoders (i.e., $\{f_e^{(j)}\}_{j \in \mathcal{X}}$) and decoders (i.e., $\{f_d^{(k)}\}_{k \in \mathcal{Y}}$). The formal definition of meta information is in Def. 2.

**Definition 2** (Meta information). Meta information of mode $k$ is $m^k$ s.t. $p(\mathcal{G}^{(k)}|m^{(k)}) > p(\mathcal{G}^{(k)})$.

When generalizing to unseen target domains, $\beta_e$ and $\beta_d$ can produce encoders and decoders of unseen modes given their meta information.

---

**Algorithm 1:** Algorithm of learning phase

**Input:** $\{\tilde{\mathcal{G}}^{(j)}, m^{(j)}\}_{j \in \mathcal{X}}$: topology, node attributes and meta information of source modes
**Input:** $\{\tilde{A}^{(k)}, m^{(k)}\}_{k \in \mathcal{Y}}$: topology and meta information of target modes
**Input:** Initialized parameters $\gamma_e$, $\gamma_d$ and $\phi$
**Output:** $\{\hat{X}_i^{(k)}\}_{k \in \mathcal{Y}}$, i=1,2,...,n

1  **while** *Converge* **do**
2      **for** $k \in \mathcal{Y}$ **do**
3          Compute attributes $\hat{\tilde{X}}_i^{(k)}$ of $\tilde{\mathcal{G}}^{(k)}$ via Eq. 7 for each sample $i$ in mode $k$
4          Assemble $\bar{A}^{(k)}$ and $\bar{X}_i^{j \to k}$ as in Eq. 10
5          Perform link prediction and update $\tilde{A}^{(k)}$ via Eq. 10
6          Compute attributes of the remaining nodes $\hat{\hat{X}}_i^{(k)}$ in $\mathcal{V} \setminus \tilde{\mathcal{V}}^{(k)}$ via Eq. 11
7          Concatenate predicted attributes of all nodes: $\hat{X}_i^{(k)} = [\hat{\hat{X}}_i^{(k)}, \hat{\tilde{X}}_i^{(k)}]$
8      Compute $\mathcal{L}_1(\gamma_e, \gamma_d)$ via Eq. 4, $\mathcal{L}_2(\phi)$ via Eq. 9 and $\mathcal{L}$ via Eq. 12
9      Update $\gamma_e$, $\gamma_d$ and $\phi$ by stochastic gradient descent on $\mathcal{L}$

---

**Learing phase**. In the training process, we learn parameters $\gamma_e$, $\gamma_d$ of the encoder hypernetwork $\beta_e$ and the decoder hypernetwork $\beta_d$, respectively, on the source domain $\mathcal{S} = \{\mathcal{X} \times \mathcal{Y} : \mathcal{X} \in \mathcal{P}(\mathcal{G}), \mathcal{Y} \in \mathcal{P}(\mathcal{G} - \mathcal{X})\}$. Specifically, we minimize the learning objective $\mathcal{L}$ of MultiHyperGNN:

$$\mathcal{L} = \mathcal{L}_1(\gamma_e, \gamma_d) + \rho \cdot \mathcal{L}_2(\phi), \tag{12}$$

where $\mathcal{L}_1$ and $\mathcal{L}_2$ are obtained from Eq. 4 and Eq. 9, respectively, $\rho$ is the hyperparameter, and $\phi$ is another trainable parateper for semi-supervised link prediction. In implementation, $\beta_e$ and $\beta_d$ are approximated by MLPs. The learning phase is also depicted in Algorithm 1.

The number of combinations for the input and output modes scales exponentially (i.e., $O(3^N)$), traditionally leading to $O(3^N)$ of computational complexity in each training epoch. By contrast, our proposed model in Algorithm 1 only needs to train on input-output mode pairs (e.g., input mode to output mode) so that we have only $O(N^2)$ such pairs to iterate in each training epoch. Therefore, the computational complexity is reduced from exponential $O(3^N)$ to polynomial $O(N^2)$.

**Generalization phase**. Once $\hat{\gamma}_e$, $\hat{\gamma}_d$ are learned as parameters of $\beta_e$ and $\beta_d$, respectively, we generalize the model to the unseen target domain $\mathcal{T} = \mathcal{X}^{\mathcal{T}} \times \mathcal{Y}^{\mathcal{T}}$ by guiding $\beta_e$ and $\beta_d$ with the meta information of unseen modes $\{m^{(j)}\}_{j \in \mathcal{X}^{\mathcal{T}}}$ and $\{m^{(k)}\}_{k \in \mathcal{Y}^{\mathcal{T}}}$. Following Eq. 1 and Eq. 3, we have:

$$\beta_e^{(j)} = \beta_e(m^{(j)}; \hat{\gamma}_e), \ j \in \mathcal{X}^{\mathcal{T}}, \quad \beta_d^{(k)} = \beta_d(m^{(k)}; \hat{\gamma}_d), \ k \in \mathcal{Y}^{\mathcal{T}}$$
$$\hat{X}_i^{(k)} = f_d^{(k)}(A^{(k)}, \sigma(\{f_e^{(j)}(G_i^{(j)}; \beta_e^{(j)})\}_{j \in \mathcal{Y}^{\mathcal{T}}}); \beta_d^{(k)}), \ k \in \mathcal{Y}^{\mathcal{T}} \tag{13}$$

We theoretically prove that in the generalization phase our model can generalize to $\mathcal{T}$ given sufficient mode-specific meta information.

**Definition 3** (Generalization error). Suppose $X_i^{(k)} = f_{\hat{\gamma}_{\mathcal{X} \to \mathcal{Y}}}(A^{(k)}, \{G_i^{(j)}, m^{(j)}\}_{j \in \mathcal{X}^{\mathcal{T}}}; \hat{\gamma}_{\mathcal{X} \to \mathcal{Y}}) + \boldsymbol{\epsilon_i}$ following Eq. 3, where $k \in \mathcal{Y}^{\mathcal{T}}$ and $\hat{\gamma}_{\mathcal{X} \to \mathcal{Y}}$ is estimated during training process on $\mathcal{S}$. We define $\|\epsilon_i\|_2^2$ as the *generalization error* of sample $i$.

**Definition 4** (Sufficient meta information). We define $m^{(j)}$ and $m^{(k)}$ as the *sufficient meta information* of the prediction $f_{\gamma_{\mathcal{X} \to \mathcal{Y}}}$ if $j \in \mathcal{X}$, $k \in \mathcal{Y}$, $m^{(j)}$ belongs to the space $\mathcal{M}$ that is a bijective mapping of the space of sufficient statistic of $\mathcal{G}^{(j)}$, and $m^{(k)} = \arg\max_m p(\mathcal{G}^{(k)}|\mathcal{G}^{(j)}, m)$.

In practice, according to Def. 2, $p(\mathcal{G}^{(k)}|m^{(k)}) = \int p(\mathcal{G}^{(k)}|\mathcal{G}^{(j)}, m^{(k)})p(\mathcal{G}^{(j)})d\mathcal{G}^{(j)} = \mathbb{E}_{p(\mathcal{G}^{(j)})}[p(\mathcal{G}^{(k)}|\mathcal{G}^{(j)}, m^{(k)})] \approx p(\mathcal{G}^{(k)}|\mathcal{G}^{(j)}, m^{(k)})$. Therefore, based on Def. 4, pursuing sufficient meta information is approximately aligned with identifying the most mode-representative one.

**Theorem 1.** *For the mode $j \in \mathcal{X}^T$ and the mode $k \in \mathcal{Y}^T$, $\mathcal{X}^{\mathcal{T}} \times \mathcal{Y}^{\mathcal{T}} \in \mathcal{T}$, $m^{(j)}$ and $m^{(k)}$ are sufficient meta information of $j$ and $k$, compute $\hat{X}_i^{(k)}$ following Eq. 3 using $\mathcal{G}^{(j)}$, $A^{(k)}$, $m^{(j)}$ and $m^{(k)}$ and calculate the generalization error $\|\epsilon_i\|_2^2$ as in Def. 3. Then compute $\{\hat{X}_i^{(k)'}\}_{k \in \mathcal{Y}^{\mathcal{T}}}$ following Eq. 3 using the same input but with $\forall m^{(j')}, j' \in \mathcal{X}, \forall m^{(k')}, k' \in \mathcal{Y}$ and $\mathcal{X} \times \mathcal{Y} \in \mathcal{S}$ as the input of $\beta_e$ and $\beta_d$. This leads to the generalization error $\|\epsilon_i'\|_2^2$. Assume $\epsilon_i$ in Def. 3 has a Gaussian distribution $\epsilon_i \sim \mathcal{N}(\mathbf{0}, \mathbf{\Sigma})$, then we have $\|\epsilon_i\|_2^2 \leq \|\epsilon_i'\|_2^2$.*

The proof of the above theory is in Appendix B. An ample amount of meta-information will result in reduced generalization error.

## 5 EXPERIMENTS

This section reports the results of experimental analysis with implementation details and complexity analysis in Appendix C and Appendix D, respectively.

### 5.1 DATASET

We evaluate on three real-world datasets: (1) **Genes**: Utilizing gene expression data from the public dataset Genotype-Tissue Expression Consortium (Lonsdale et al., 2013), we focus on five tissues: whole blood (WB), lung (L), muscle skeletal (MS), sun-exposed (LG), and not-sun-exposed skin (S). The gene-gene network is formed via weighted correlation network analysis (Langfelder & Horvath, 2008) with expression values as node attributes. Meta-information comprises tissue type, location, structure, function, and cell types; (2) **Climate**: Derived from the Goddard Earth Observing System Composition Forecasting (2019-2021), air temperature data for US state capitals is segmented into four daily modes: early morning (0:00AM-6:00AM), late morning (6:00AM-12:00PM), afternoon (12:00PM-18:00PM), and night (18:00PM-0:00AM). Cities represent graph nodes with air temperature as node attributes, connected based on high Pearson Correlation of temperatures. Time period indicators and timestamps serve as meta-information; (3) **Traffic**: Employing the PEMS08 dataset, we assess domain generalization in San Bernardino traffic data (July-August 2016), with 170 detectors across eight roads. Meta-information is encoded as one-hot vectors for speed, occupancy, and flow, with all measures min-max normalized. Dataset details are provided in Appendix E.

### 5.2 EVALUATION METRICS

We evaluate the model performance both quantitatively and qualitatively. For quantitative evaluation, we measure prediction accuracy based on MSE and Pearson Correlation Coefficients (PCC). To evaluate the efficiency, we theoretically analyze the space complexity of MultiHyperGNN and other models. For qualitative evaluation, we visualize the distribution between predicted and ground-truth node attributes in unseen modes during the testing process.

### 5.3 COMPETING MODELS AND ABLATION STUDIES

We employ five competing models: (1) **ED-GNN**. We modify MultiHyperGNN to a naive encoder-decoder-based model by directly training the encoder and the decoder for each mode combination. A single model is trained for all mode combinations; (2) **Multi-Head Model (MHM)** (Vandenhende et al., 2021). We modify ED-GNN into a multi-task learning framework by simultaneously training multiple decoders with the same encoder. This model can only be used for prediction instead of domain generalization since each decoder deals with a specific output mode; (3) **Interaction Networks (IN)** (Battaglia et al., 2016). IN models interactions and dynamics of nodes in the graph for node-level graph transformation. Particularly, IN uses only fixed graph topology from the input mode; (4) **Explore-to-Extrapolate Risk Minimization (EERM)** (Wu et al., 2022). EERM employs $q$ context explorers that undergo adversarial training to maximize the variance of risks across multiple virtual environments. This enables the model to extrapolate from a single observed

Table 1: Evaluation on prediction accuracy.

| Model | Genes-L | | Genes-LG | | Genes-S | | T-Afternoon | | T-Night | |
|---|---|---|---|---|---|---|---|---|---|---|
| | MSE | PCC | MSE | PCC | MSE | PCC | MSE | PCC | MSE | PCC |
| ED-GNN | 1.9810 | 0.6072 | 2.1289 | 0.5795 | 2.1925 | 0.5764 | 59.3010 | 0.4539 | 84.0824 | 0.4187 |
| MHM | 2.0126 | 0.5913 | 2.0153 | 0.5312 | 2.0384 | 0.5816 | 61.2798 | 0.4300 | 69.8599 | 0.4207 |
| IN | 2.0182 | 0.6026 | 2.2019 | 0.5683 | 2.1304 | 0.5377 | 60.8755 | 0.4650 | 71.0456 | 0.4210 |
| EERM | 1.8624 | 0.6493 | 1.9035 | 0.6325 | 2.1187 | 0.5931 | 84.0604 | 0.4259 | 83.2518 | 0.4101 |
| DRAIN | 1.9798 | 0.6132 | 1.9969 | 0.6009 | 2.2100 | 0.5741 | 91.4561 | 0.3987 | 104.3200 | 0.4085 |
| HyperGNN-1 | 2.7566 | 0.2574 | 2.8543 | 0.2494 | 2.8863 | 0.2501 | 129.6152 | 0.3566 | 101.0478 | 0.4095 |
| HyperGNN-2 | 2.9383 | 0.2654 | 3.0230 | 0.2565 | 3.0467 | 0.2574 | 280.5912 | 0.3557 | 400.0514 | 0.3125 |
| HyperGNN | 1.9720 | 0.6144 | 2.2040 | 0.5700 | 2.1930 | 0.5799 | 69.3157 | 0.4405 | 70.0319 | 0.4299 |
| MultiHyperGNN-MLP | 2.8958 | 0.2608 | 3.4251 | 0.2736 | 3.6073 | 0.2814 | 104.0525 | 0.3764 | 81.9324 | 0.4122 |
| MultiHyperGNN-S | 2.0023 | 0.6492 | 2.2420 | 0.6018 | 2.2723 | 0.6153 | 89.1604 | 0.4027 | 75.6518 | 0.4151 |
| MultiHyperGNN-GCN | 1.8023 | 0.6511 | 1.9426 | 0.6340 | 1.9539 | 0.6337 | 89.1321 | 0.4395 | 68.7137 | 0.4216 |
| MultiHyperGNN-GIN | **1.7101** | **0.6654** | **1.8913** | **0.6450** | 1.9046 | 0.6455 | **43.5142** | **0.5155** | **49.0168** | **0.4878** |
| MultiHyperGNN-GAT | 1.7695 | 0.6583 | 1.9107 | 0.6347 | **1.8951** | **0.6470** | 54.2913 | 0.4937 | 60.1922 | 0.4561 |

Table 2: Evaluation on domain-generalization accuracy.

| Model | Genes-L | | Genes-LG | | Genes-S | | Traffic-Flow | | Traffic-Speed | |
|---|---|---|---|---|---|---|---|---|---|---|
| | MSE | PCC | MSE | PCC | MSE | PCC | MSE | PCC | MSE | PCC |
| ED-GNN | 2.2387 | 0.4752 | 2.0573 | 0.5229 | 2.0425 | 0.5511 | 1.3070 | 0.5316 | 1.2108 | 0.5566 |
| IN | 2.1017 | 0.5312 | 2.1539 | 0.5249 | 2.3746 | 0.4795 | 1.0914 | 0.5961 | 1.1679 | 0.6019 |
| EERM | 2.2148 | 0.5193 | 2.3536 | 0.4583 | 2.5792 | 0.4669 | 1.2127 | 0.5663 | 1.2015 | 0.5690 |
| DRAIN | 2.8155 | 0.5123 | 3.2461 | 0.4016 | 3.2777 | 0.4230 | 1.1195 | 0.5906 | 1.1698 | 0.5919 |
| HyperGNN-1 | 3.7586 | 0.2359 | 3.3152 | 0.2614 | 3.3011 | 0.2537 | 1.2055 | 0.5438 | 1.2017 | 0.5788 |
| HyperGNN-2 | 3.1516 | 0.2338 | 3.3064 | 0.2572 | 3.5869 | 0.2629 | 1.1984 | 0.5526 | 1.2911 | 0.5209 |
| HyperGNN | 1.9025 | 0.6003 | 2.0471 | 0.6427 | 1.9913 | 0.6236 | 1.0698 | 0.6002 | 1.0980 | 0.6125 |
| MultiHyperGNN-MLP | 3.0812 | 0.2150 | 3.1519 | 0.2963 | 3.6322 | 0.3049 | 1.1518 | 0.5834 | 1.2080 | 0.5857 |
| MultiHyperGNN-GCN | 1.8513 | 0.6495 | 2.0086 | 0.6410 | 1.9965 | 0.6127 | 1.0615 | 0.6164 | 1.1330 | 0.6087 |
| MultiHyperGNN-GIN | **1.8005** | **0.6600** | **1.9852** | **0.6479** | 1.9031 | **0.6471** | **0.9055** | **0.6312** | **1.0418** | **0.6309** |
| MultiHyperGNN-GAT | 1.8069 | 0.6562 | 2.0123 | 0.6455 | **1.8921** | 0.6425 | 0.9946 | 0.6253 | 1.1011 | 0.6102 |

environment; (4) **DRAIN** (Bai et al., 2022). DRAIN utilizes recurrent graph generation to generate dynamic graph-structured neural networks using hypernetworks trained on various time points. This framework can capture the temporal drift of both model parameters and data distributions to make future predictions. Additionally, we modify MultiHyperGNN to evaluate four different aspects: (1) **HyperGNN**. HyperGNN is a simpler version of MultiHyperGNN by predicting multiple output modes from one single input mode. In this case, only $\beta_d$ is trained; (2) **HyperGNN-1**. To explore if a single MLP prediction layer can predict for all output modes, we will not produce MLP layers but only produce GAT layers by hypernetwork; (3) **HyperGNN-2**. In our experimental setting the meta information is composed of mode types (one-hot vector) and other mode-related features. For HyperGNN-2, we reduce the meta information by only feeding the mode type to hypernetworks; (4) **MultiHyperGNN-S**. Graph transformation from multiple input modes is expected to power the prediction by aggregating from these input modes. To validate this assumption, during the testing process of MultiHyperGNN, we will not use only a single input mode as the input data.

## 5.4 QUANTITATIVE EVALUATION

**Evaluating prediction accuracy**. On the Genes dataset, we train MultiHyperGNN to predict gene expression in various tissues using data from whole blood and muscle skeletal. For testing MultiHyperGNN-S, we use only whole blood data. HyperGNN and its variants are trained similarly but with whole blood as the sole input. EERM and DRAIN are also trained from a single mode. In the Climate dataset, we train MultiHyperGNN using early and late morning air temperatures to predict afternoon and night temperatures. HyperGNN and its variants, EERM, and DRAIN are trained from late morning data only. ED-GNN and IN are trained on all input-output mode combinations, while MHM follows the same training strategy as HyperGNN.

As shown in Table 1, MultiHyperGNN achieves superior performance on both datasets. The MSE of MultiHyperGNN-GIN is 0.1262 (6.43%) smaller than the second best model, EERM, by average. The PCC of MultiHyperGNN-GIN is 0.0270 (4.32%) higher than the second best model, EERM, by average. This is expected since MultiHyperGNN involves two input modes so that it is more expressive than EERM. HyperGNN, MultiHyperGNN-S and other competing models have comparable results since they all predict from a single input mode. Compared with HyperGNN, the performance of HyperGNN-1 is worse, indicating that a mode-specific prediction layer is still needed. In addition, the deployment of MultiHyperGNN hinges upon the accessibility of mode-specific meta-information. As evidenced in Table 1 and Table 2, the utilization of HyperGNN-2, which condenses meta-information to only the mode type, results in suboptimal prediction accuracy across almost all settings.

**Evaluating domain generalization**. We evaluate MultiHyperGNN and other models regarding domain generalization using Genes and Traffic dataset. To evaluate the generalization ability on a specific output mode (e.g., Genes-L or Traffic-Flow), each time we train the model to predict another modes (e.g., Genes-LG, Genes-S or Traffic-Speed) using data of whole blood and muscle skeletal in Genes or occupancy in Traffic as input modes. During testing time, we apply the trained model to the output mode (e.g., Genes-L or Traffic-Flow) and calculate the prediction accuracy.

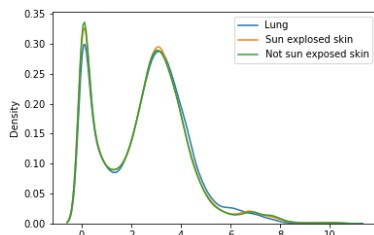

As per Table 2, MultiHyperGNNs shows consistently better performance compared with other models. Notably, MultiHyperGNN-GIN exhibits an average MSE of 0.0840, which is 4.24% lower than the second-best model, HyperGNN, with an average MSE of 1.9803. MultiHyperGNN-GIN has the PCC 0.0295 (4.74%) higher by average than the second model, HyperGNN, which has the PCC of 0.6222 by average. In the experiment predicting highway traffic flow and speed, MultiHyperGNN-GIN surpasses the second-best model by 7.62% and 1.97% on average in MSE and PCC, respectively, for domain generalization. The better performance of Multi-HyperGNN results from the fact that MultiHyperGNN predicts

Figure 3: Density plot of gene expression in lung, sun-exposed and not-sun-exposed skin in testing data.

from multiple input modes, which is more expressive than HyperGNN that only achieves single-input, multi-output mode prediction. The superior performance of MultiHyperGNN and HyperGNN compared with other models results from the meta information that guides hypernetworks to generalize the model to unseen domains.

## 5.5 QUALITATIVE EVALUATION

We visualize the distribution of node attributes in different modes of Genes. As shown in Figure 5.4, in the testing data the distribution of sun-exposed skin is similar to the not-sun-exposed skin. This is reasonable since both are skin tissues and they share similar meta information. By contrast, lung is different from skin, so that its distribution is different from two skin tissues. This also confirms the necessity to design the model to handle mode similarities. We also visualize via density plots the alignment of the distribution of predicted values with the ground-truth distribution in unseen testing data (Figure 4) corresponding to the results in Table 2. Based on the results in Figure 4, in all three human tissues, MultiHyperGNN achieves roughly the same distribution with the ground-truth distribution of the testing data, which is much better than other competing models. This is aligned with the superior prediction accuracy in domain generation of MultiHyperGNN as shown in Table 2.

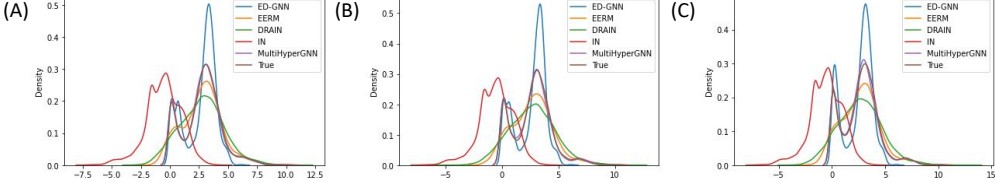

Figure 4: Density plots to visualize the distribution of predicted and ground-truth gene expression in testing data, including density plot for (A) lung, (B) sun-exposed skin and (C) not-sun-exposed skin.

## 6 CONCLUSION

In this paper, we address challenges in domain generalization deep graph transformation. Firstly, we pinpoint three challenges in domain generalization graph generalization, followed by introducing MultiHyperGNN with an encoder and decoder to encode graph topologies in input and output modes. Two novel hypernetworks are designed to generate the encoder and decoder, steered by mode-specific meta-information for domain generalization. Experiments demonstrate superior performance of MultiHyperGNN over competing models. Future work includes investigating key meta-information components to optimize model performance. Code for result reproduction is provided in the supplemental materials.

# 7 ETHICAL STATEMENT

We develop our method from publicly available GTEx and PEMS08 datasets. Climite dataset is curated from public Goddard Earth Observing System Composition Forecasting (2019-2021). Note that for gene expression data in GTEx, we only access public normalized gene expression of the dataset without any private information of samples. The model is only able to predict normalized cross-tissue gene expression, not the true expression.

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

# A    NOTATION TABLE

A detailed notation table is in Table 3.

# B    PROOF OF THEOREM 1

Consider the situation that we train $f_{\gamma_{\mathcal{X} \to \mathcal{Y}}}$ (Eq. 3), parameterized by $\gamma_{\mathcal{X} \to \mathcal{Y}}$, on $\mathcal{S}$ s.t. $\mathcal{X} \times \mathcal{Y} \in \mathcal{S}$. Without loss of generalization, in the target domain, we consider predicting node attributes in mode $k$ from mode $j$, where $j \in \mathcal{X}^{\mathcal{T}}$ and $k \in \mathcal{Y}^{\mathcal{T}}$. First of all, we explain that solving the following conditional likelihood is equivalent to minimizing generalization error:

$$m^* = \arg \max_m p(\mathcal{G}^{(k)}|\mathcal{G}^{(j)}, m), \quad j \in \mathcal{X}^{\mathcal{T}}, \quad k \in \mathcal{Y}^{\mathcal{T}}, \tag{14}$$

where $m = (m^{(j)'}, m^{(k)'})$ and $m^* = (m^{(j)*}, m^{(k)*})$, $\mathcal{G}^{(j)} = \{G_1^{(j)}, G_2^{(j)}, ..., G_n^{(j)}\}$ and $\mathcal{G}^{(k)} = \{G_1^{(k)}, G_2^{(k)}, ..., G_n^{(k)}\}$. Then, we prove that $m^*$ that satisfies Eq. 14 should be sufficient meta information of mode $j \in \mathcal{X}^{\mathcal{T}}$ and $k \in \mathcal{Y}^{\mathcal{T}}$.

## B.1    SOLVING EQUATION 14 IS EQUIVALENT TO MINIMIZING GENERALIZATION ERROR

Since $X_i^{(k)} = f_{\hat{\gamma}_{\mathcal{X} \to \mathcal{Y}}}(A^{(k)}, \{G_i^{(j)}, m^{(j)}\}_{j \in \mathcal{X}^{\mathcal{T}}}, m^{(k)}; \hat{\gamma}_{\mathcal{X} \to \mathcal{Y}}) + \epsilon_i = \hat{X}_i^{(k)} + \epsilon_i$ where $\hat{\gamma}_{\mathcal{X} \to \mathcal{Y}}$ is learned and fixed, considering $m = \{m^{(j)'}, m^{(k)'}\}$ as the random variable, we have $\hat{X}_i^{(k)} - X_i^{(k)} = \epsilon_i \sim \mathcal{N}(\mathbf{0}, \mathbf{\Sigma})$. As a result, take a logarithm of $p(\mathcal{G}^{(k)}|\mathcal{G}^{(j)}, m)$, we have:

$$\begin{aligned} \log p(\mathcal{G}^{(k)}|\mathcal{G}^{(j)}, m) &= \log p(\{X_i^{(k)}\}_{i=1}^n, A^{(k)}|\{G_i^{(j)}\}_{i=1}^n, m) = \\ &= \log p(\{X_i^{(k)}\}_{i=1}^n|A^{(k)}, \{G_i^{(j)}\}_{i=1}^n, m) \\ &+ \log p(A^{(k)}|\{G_i^{(j)}\}_{i=1}^n, m) \end{aligned} \tag{15}$$

Since $A^{(k)}$ is known, then maximizing Eq. 14 is equivalent to maximizing the first term of the above equation. Given $n$ independent samples in the dataset:

$$\begin{aligned} m^* &= \arg \max_m \log p(\{X_i^{(k)}\}_{i=1}^n|A^{(k)}, \{G_i^{(j)}\}_{i=1}^n, m) \\ &= \arg \max_m -\tfrac{1}{2} \sum_{i=1}^n (\hat{X}_i^{(k)} - X_i^{(k)})^T \mathbf{\Sigma}^{-1}(\hat{X}_i^{(k)} - X_i^{(k)}) + C, \end{aligned} \tag{16}$$

where $C$ is a constant. As the covariance matrix $\mathbf{\Sigma}$ is positive semidefinite, the above objective is equivalent to minimizing the generalization error $\|\epsilon_i\|_2^2 = \|\hat{X}_i^{(k)} - X_i^{(k)}\|_2^2$.

## B.2    THE SUFFICIENT META INFORMATION OF $j \in \mathcal{X}^{\mathcal{T}}$ AND $k \in \mathcal{Y}^{\mathcal{T}}$ SATISFIES EQUATION 14

Based on the Bayes' theorem, we have $p(\mathcal{G}^{(j)}, \mathcal{G}^{(k)}|, m^{(j)'}, m^{(k)'}) = p(\mathcal{G}^{(k)}|\mathcal{G}^{(j)}, m^{(j)'}) \cdot p(\mathcal{G}^{(j)}|m^{(j)'})$. If $m^{(j)'}$ and $m^{(k)'}$ are sufficient meta information $m^{(j)}$ and $m^{(k)}$. Therefore, $p(\mathcal{G}^{(j)}|m^{(j)'}) = 1$ so that we have $p(\mathcal{G}^{(j)}, \mathcal{G}^{(k)}|, m^{(j)'}, m^{(k)'}) = p(\mathcal{G}^{(k)}|\mathcal{G}^{(j)}, m^{(j)'})$. Also, we have $m^{(k)} = \arg \max_m p(\mathcal{G}^{(k)}|\mathcal{G}^{(j)}, m)$, equivalently, $p(\mathcal{G}^{(j)}, \mathcal{G}^{(k)}|, m^{(j)'}, m^{(k)'})$ is maximized by $m^{(k)}$. In conclusion, the sufficient meta information of $j \in \mathcal{X}^{\mathcal{T}}$ and $k \in \mathcal{Y}^{\mathcal{T}}$ satisfies Eq. 14.

# C    COMPLEXITY ANALYSIS

We compare MultiHyperGNN with other models by the theoretical space complexity analysis. To train a predictive mapping that covers all mode combinations in $\mathcal{S}$, ED-GNN, MHM and IN have $O(3^N)$ encoders and decoders in total that need to be trained, leading to $O(3^N)$ space complexity. EERM requires to train $q$ generators whereas DRAIN only needs a hypernetwork to produces classifiers at each time point. Therefore, EERM and DRAIN have the space complexity of $O(p)$ and $O(1)$, respectively. To train MultiHyperGNN, instead, we only need to train two hypernetworks, whose space complexity is $O(1)$ which is much smaller than competing models except DRAIN.

Regarding computational complexity, similar to space complexity, ED-GNN, MHM and IN have $O(3^N)$ mode combinations to iterate in each training epoch. The computational complexity of EERN and DRAIN is linear to the number of generators and the number of combinations of time points, which are usually much larger than $N^2$. MultiHyperGNN only needs to train on different input-output mode pairs so that we have only $O(N^2)$ such pairs to iterate in each training epoch, achieving a polynomial computational complexity.

Table 3: Table for notations

| Notation | Description |
|---|---|
| $N$ | Total number of modes in the dataset |
| $\mathcal{G}^{(j)}$ | Graph of the mode $j$ |
| $G_i^{(j)}$ | Graph of sample $i$ in the mode $j$ |
| $A^{(j)}$ | Adjacency matrix of mode $j$ |
| $X_i^{(j)}$ | Node attributes of sample $i$ in the mode $j$ |
| $\mathcal{X}$ | Set of input modes |
| $\mathcal{Y}$ | Set of output modes |
| $\mathcal{S}$ | Source domain |
| $\mathcal{X}^{\mathcal{T}}$ | Set of input modes in the target domain |
| $\mathcal{Y}^{\mathcal{T}}$ | Set of output modes in the target domain |
| $\mathcal{T}$ | Target domain |
| $f$ | Graph transformation function |
| $f_e^{(j)}$ | Encoder to encode graph of mode $j$, parameterized by $\beta_e^{(j)}$ |
| $f_d^{(k)}$ | Decoder to decoder graph of mode $k$, parameterized by $\beta_d^{(k)}$ |
| $\beta_e$ | Encoder hypernetwork to produce $\{f_e^{(j)}\}_{j \in \mathcal{X}}$, parameterized by $\gamma_e$ |
| $\beta_d$ | Decoder hypernetwork to produce $\{f_d^{(k)}\}_{k \in \mathcal{Y}}$, parameterized by $\gamma_d$ |
| $m^{(j)}$ | Meta information of mode $j$ |

## D  IMPLEMENTATION DETAILS

All experiments are conducted by Python 3.9 on the 64-bit machine with an NVIDIA GPU, NVIDIA GeForce RTX 3090. In practice we use six-layer MLPs with the hidden dimension as 7200 to model $\gamma_e$ and $\gamma_d$. The source encoder and the target decoder contain two-layer GNNs (with four heads if GATs) and the hidden dimension of 256. The prediction layer is composed of five-layer MLPs with the hidden dimension of 512.

## E  STATISTICS OF DATASETS IN EXPERIMENTS

**Genes**. We use gene expression data collected and curated by the Genotype-Tissue Expression (GTEx) Consortium (Lonsdale et al., 2013). Specifically, processed gene expression data derived from bulk RNA-seq experiments on five tissues, whole blood (WB), lung (L), muscle skeletal (MS), sun-exposed skin (lower leg, LG), not-sun-exposed skin (suprapubic, S) are used. For quality control purpose, we first remove samples with no data in any of the five tissues types. Then we remove genes with low expression level (total number of mapped reads less than 2 across all samples). For the remaining data, for each tissue, we perform weighted correlation network analysis (WGCNA) (Langfelder & Horvath, 2008) with the cutoff $\rho$ on the correlation coefficients to construct the co-expression network with the expression value as the node attribute. The meta information that characterizes these five tissues includes tissue type (lung, muscle, skin), location (trunk, leg, arm), structure (dense, rigid, spongy), function (movement, protection, gas exchange) and cell types (alveoli and bronchioles, cylindrical muscle fibers, epithelial cells).

**Climate**. We use the Goddard Earth Observing System Composition Forecasting (GEOS-CF) hourly historical meteorological data[1] across the contiguous United States from 2019-2021. The GEOS-CF meteorological data is assimilated from a variety of conventional and satellite-driven data sources. The detailed assimilation approaches can be found at the website[2]. We further calculate the surface air temperature (T) for each state capital by averaging data from all the GEOS-CF pixels that fall within a given capital city plus a 10 km buffer region. Specifically, we split 24 hours of a day into four time periods as four modes: early morning (0:00AM-6:00AM), late morning (6:00AM-12:00PM), afternoon (12:00PM-18:00PM) and night (18:00PM-0:00AM), and calculate the mean value of the air temperature in each period. To construct the network in each domain, we use cities as graph

---

[1] https://gmao.gsfc.nasa.gov/weather_prediction/GEOS-CF/
[2] https://ntrs.nasa.gov/citations/20120011955

Table 4: Statistics of employed datasets, Genes and Climate ($|\mathcal{D}|$ is the number of domains contained in the dataset; $p$ is the number of nodes; $|\mathcal{E}|_{avg}$ is the average number of edges of the graph in the dataset by domain; $n$ is the number of samples in each dataset).

| Model | Genes | Climate | Traffic |
|---|---|---|---|
| $|\mathcal{D}|$ | 5 | 4 | 4 |
| $p$ | 2,398 | 48 | 170 |
| $|\mathcal{E}|_{avg}$ | 28477/1174/1020/7463/272 | 293/266/336/272 | 227/227/227/227/227 |
| $n$ | 205 | 1,095 | 17,833 |

nodes and air temperature in each city as the node attribute. Then in each time period, we calculate the correlation of the air temperature between two cities within three years. If the correlation is greater than $\rho$ then there is an edge connecting two cities on the graph. We use the time period indicator (four-element, one-hot vector to indicate early morning, late morning, afternoon and night) and various time stamps when the data is collected as the meta information.

The statistics of these two datasets are in Table 4. The data is split as 80% training, 10% testing, and 10% validation.

