# OpenReview forum: "Domain Generalization Deep Graph Transformation"
_ICLR.cc/2024/Conference — Submitted to ICLR 2024_

### Official Review · Reviewer_oPHk · 2023-10-26

**Soundness:** 3 good
**Presentation:** 2 fair
**Contribution:** 3 good
**Rating:** 5
**Confidence:** 3

**Summary:**

The presented work proposes a novel model that learns to solve graph transformation problem under a domain generalization setting.

**Strengths:**

- The superiority of the proposed framework is clearly stated and tested through experiments.
- The proposed model targets on domain generalization, which is a major problem in graph representation learning.
- The use of hypernetworks in graph transformation is novel, as far as I know.
-----
after reading the author responce, I decide to keep my current rating to this paper unchanged.

**Weaknesses:**

- I find this paper vague in some places with its wording. See Questions.
- I don't quite understand what is the essence of the optimization of space complexity in this model. The authors claim that, if we traverse all possible combinations of input and output pairs it will require $\Omega(3^N)$ time (BTW the paper uses the asymptotic notation $O$, but actually should be $\Omega$ here), which is true, but usually you don't require to traverse all possible input-output pairs to learn a not-bad model, just like in ordinary supervised classification, the model doesn't need to travers all possible samples from the ditribuition but just a finite subset of them (the training set). In my understanding, the presented paper basically asumed there is a distribution of meta information and the training of the proposed model will require meta information as input. However, if there is indeed an underlying distribution of meta information, then just simply sampling some pairs from all the $\Omega(3^N)$ possible combination might also work well. That's why I don't quite understand the necessity of using this multi-input, multi-output, hypernetwork-based framework here. It is possible that my understanding is wrong. If so feel free to point it out.
- I don't understand why the authors claim the space complexity of the propsed model is constant. Based on Algorithm 1, the output size  is linear in the size of $\mathcal Y$ and $n$.
- In Theorem 1, it makes no sense to me to assume the genralization error to be iid Gaussian (as the notation is quite confusing here, I don't actually know what does the authors mean here, but I guess it means iid Gaussian. Please correct me if my understanding is wrong). According to Definition 3, the error comes from $X_i ^{(k)} - f_{\hat \gamma_{\mathcal X\to \mathcal Y}}(\cdots)$, and as $f_{\hat \gamma_{\mathcal X\to \mathcal Y}}$ is based on GNNs, there should be inter-node interactions. It is very unlikely ${\boldsymbol{\epsilon}}_i$ can be iid.

**Questions:**

- In Section 3, there's a sentence "where $A^{(j)} \in \mathbb R^{p_j \times p_j}$ is the graph of size $p_j \leq p$". Do you mean $A^{(j)}$ is the **adjancency** of a graph? Also, sincein the definition $A^{(j)} \in \mathbb R^{p_j \times p_j}$, do you mean the entries of $A^{(j)}$ can be take any real numbers, even negative numbers?
- In Definition 1, what does "s.t. $\mathcal X^{\mathcal  T} \times \mathcal Y^{\mathcal T} \not \in \mathcal S$" mean?  Does that mean, say $\mathcal X^{\mathcal T}$ contains modes that are not in $\mathcal G$?
- In equation (1), what is $\beta_d^{(k)}$? What is the dimensioanlity of $\hat X_i^{(k)}$?
- In the definition of $f_{\gamma_{\mathcal X \to \mathcal Y}} = \left\\{f_d^{(k)} \star \left\\{f_e^{(j)}\right\\}\_{j \in \mathcal X}\right\\}_{k \in \mathcal Y}$, what does $\star$ mean here?
- In Algorithm 1, what is $i$ in the output? It doesn't seems to be in the input nor is it a loop variable.
- In Theorem 1, ${\boldsymbol{\epsilon}}_i \sim \mathcal N({\boldsymbol{0}}, {\boldsymbol{\sigma}}^2)$, what does ${\boldsymbol{\sigma}}^2$ mean? Do you mean $\sigma^2 {\boldsymbol{I}}$ where ${\boldsymbol{I}}$ is the identity matrix?

---

> ### Author Response · Authors · 2023-11-16
> **Response to reviewer oPHk**
>
> We greatly value the reviewer's feedback and offer the following clarifications.
>
> **Comment 1:** Usually you don't require to traverse all possible input-output pairs to learn a not-bad model, just like in ordinary supervised classification, the model doesn't need to travers all possible samples from the ditribuition but just a finite subset of them (the training set). In my understanding, the presented paper basically assumed there is a distribution of meta information and the training of the proposed model will require meta information as input. However, if there is indeed an underlying distribution of meta information, then just simply sampling some pairs from all the possible combination might also work well.
>
> **Response 1**:
>
> * It is possible to assign distinct encoders and decoders for each input and output mode, respectively, and sample a subset of input-output pairs to bypass training on all possible input-output combinations. This approach ensures the inclusion of all modes during training. Our baseline model, ED-GNN, employs this method. It demonstrates comparable performance to MultiHyperGNN in prediction tasks (refer to Table 1), but exhibits lower domain generalization performance (see Table 2). This limitation arises due to the absence of meta-information guiding the generalization process.
>
> * Alternatively, we can indeed use a singular encoder for all input modes and a single decoder for all output modes. In the absence of meta-information, this approach does not reveal relationships among modes. This is crucial as the same input graph may predict different output graphs under varying contexts. By contrast, utilizing meta-information, however, poses two training challenges:
>
>     - Encode the graph into graph embedding and concatenate the meta information into the embedding, but then we cannot enjoy node level embedding for downstream prediction task at the node level.
>     - Duplicating meta-information for each node to merge with node embeddings leads to redundancy. Additionally, this does not effectively capture the dependency of node inputs on meta-information due to the nature of concatenation.
>
> * Contrary to the aforementioned approaches, our method efficiently handles all input-output pairs without resorting to subset sampling. This leads to more precise results as we could involve all input and output modes, but at a cost of a much smaller trainable parameter space.
>
> We believe these clarifications underscore the effectiveness and innovation of our approach in comparison to existing methodologies.
>
> **Comment 2:** I don't understand why the authors claim the space complexity of the propsed model is constant. Based on Algorithm 1, the output size is linear in the size of $\mathcal Y$ and $n$.
>
> **Response 2:**
>
> * The space complexity in our discussion is defined in terms of the total number of trainable parameters within the model. Traditional methods typically require training an exponential number of encoder-decoder combinations, approximately $O(3^N)$ where $N$ is total number modes in dataset, to manage the exponential variety of source-target mode combinations. This requirement results in an exponential space complexity of trainable parameters.
>
> * In contrast, our proposed model significantly simplifies this aspect. It involves training only two hypernetworks, regardless of the number of mode combinations. This approach inherently limits the space complexity to a constant level. We have further clarified this in the first bullet point of our contributions (highlighted in red in revised paper), stating:
>
>     “We introduced a novel framework leveraging a multi-input, multi-output training strategy, significantly reducing the space complexity regarding trainable parameters from exponential to constant during training.”

---

> ### Author Response · Authors · 2023-11-16
> **Further response to reviewer oPHk**
>
> **Comment 3:** According to Definition 3, the error comes from $X_i^{(k)}-f_{\hat{\gamma}_{\mathcal X-\mathcal Y}}$, and as it is based on GNNs, there should be inter-node interactions. It is very unlikely $\mathbf\epsilon_i$ can be iid.
>
> **Response 3:**
>
> * Our model assumes that samples (i.e., graphs) within the dataset are independent and identically distributed (iid), as opposed to individual nodes in a graph.
>
> * Notation Explanation: In our notation, $G_i^{(j)}$​ denotes the graph corresponding to the $i$-th sample in mode $j$. Therefore, the generalization error is a vector with the length of the number of nodes in the graph. We concur with the reviewer's point that the covariance matrix of the generalization error, \epsilon, is not diagonal. This acknowledgment is indeed compatible with our theoretical framework and its proofs. To enhance clarity and prevent any misunderstanding: We have revised our notation of this covariance matrix to $\Sigma$. Accordingly, we have updated Equation (16) in our manuscript to reflect this change in notation:
>
> "$m^* = argmax_m -\frac{1}{2}\sum_{i=1}^n(\hat X_i^{(k)}-X_i^{(k)})^T\mathbf\Sigma^{-1}(\hat X_i^{(k)}-X_i^{(k)}) + C$" and also:
>
> " As the covariance matrix $\mathbf\Sigma$ is positive semidefinite, the above objective is equivalent to minimizing the generalization error $\|\mathbf\epsilon_i\|_2^2=\|\hat X_i^{(k)}-X_i^{(k)}\|_2^2$"
>
> **Comment 4:** In Section 3, there's a sentence "where $A^{(j)} \in \mathbb R^{p_j \times p_j}$ is the graph of size ". Do you mean $A^{(j)}$ is the adjancency of a graph? Also, since in the definition, do you mean the entries of $A^{(j)}$ can be take any real numbers, even negative numbers?
>
> **Response 4:** Thank you for your inquiry about our notation. We confirm that $A^{(j)}$ represents the adjacency matrix of the graph in mode j. In this matrix, the elements are binary, with a value of 1 indicating the presence of an edge between nodes, and a value of 0 indicating no edge. We have clarified this in the revised paper:
>
> "$A^{(j)}\in\mathbb{R}^{p_j\times p_j}$ is the adjacency matrix of size $p_j\le p$"
>
> **Comment 5:** In Definition 1, what does "s.t. $\mathcal X^{\mathcal T} \times \mathcal Y^{\mathcal T} \not \in \mathcal S$" mean? Does that mean, say contains modes that are not in $\mathcal G$?
>
> **Response 5:** Yes, this means that target domain can be exclusive to source domain.
>
> **Comment 6:** In equation (1), what is $\beta_d^{(k)}$? What is the dimensioanlity of $X_i^{(k)}$?
>
> **Response 6:**
>
> * $\beta_d^{(k)}$ is parameter of the decoder of mode $k$ generated by the decoder hypernetwork $\beta_d$. We have highlighted in red the Eq. (2) in the revised paper that illustrates how they are computed.
>
> * Dimensionality of $\hat{X_i}^{(k)}$ is $p$, which involves all nodes (the union of input and output graphs).
>
> **Comment 7:** In the definition of $f_{\gamma_{\mathcal X \to \mathcal Y}} =${$f_d^{(k)} \star${$f_e^{(j)}$}${j \in \mathcal X}$}${k \in \mathcal Y}$, what does $\star$ mean here?
>
> **Response 7:** We appreciate the valuable input from the reviewer and giving us the opportunity to enhance our paper. $\star$ means the function convolution that encode nodes with $f_e^{(i)}$ and decode it with $f_d^{(k)}$.
>
> **Comment 8:** In Algorithm 1, what is in the output? It doesn't seems to be in the input nor is it a loop variable.
>
> **Response 8:** We have explained the output in the 4-th line of Algorithm 1: {$\hat X_i^{(k)}$}$_{k\in\mathcal{Y}}$. We have also highlighted it in red for the reviewer to review.
>
> **Comment 9:** In Theorem 1, what $\mathbf\sigma^2$ does mean? Do you mean where is the identity matrix?
>
> $\mathbf\sigma^2$ is the covariance matrix of generalization error, which is not a diagonal matrix due to the potential correlation among nodes. To clarify this, we have modified it into $\mathbf\Sigma$.

---

> ### Comment · Reviewer_oPHk · 2023-11-22
>
> I thank the authors for their responce. I decide to keep my current rating to this paper unchanged, mainly because there are too many confuding mathematical notations. I would also like to indicate that many of my concerns are not addressed.
> - For the eq.(16), the authors wrote in the revised version "as the covariance matrix is PSD, the objective is equivalent to minimizing the generalization error $\\|\epsilon\\|_2^2 = \\|\hat X_i^{(k)} - X_i^{(k)}\\|_2^2$". This is obviously not true.
> - For the constant space complexity. I understand the space complexity of proposed model is much lower than $3^N$, but it doesn't look like constant. In my understanding at least you need to store all $X_i$-s, and that is already linear space complexity.

---

> > ### Author Response · Authors · 2023-11-23
> > **Further response to reviewer oPHk**
> >
> > We sincerely appreciate the reviewer's comments for enhancing our paper. Below is our clarification to further address reviewer's concerns:
> >
> > * Complex notation: for further clarification on the notation, we have added a notation table in Appendix Table 3 as a reference for our readers. Hope this will address review's concern.
> >
> > * We sincerely appreciate reviewer's comments to identify the typo in our paper. We assume $\Sigma$ is positive definite to make it an invertible matrix in Eq. (16). We have corrected this in the revision of our paper.
> >
> > * We appreciate the reviewer's comments. Suppose $X_i\in\mathbb R^{p}$, then we agree with the reviewer that the complexity is linear to $p$, but this also applies to traditional methods which also have prediction layers and, in this case, may have the space complexity of $O(p3^N)$. Additionally, in graph transforma task, the dimension of final outputs is always a fixed number. For instance, the number of genes in human genome is always fixed. Therefore, to compare space complexity, we only focus on the number of set of parameters to be trained and this won't affect a fair comparison between our method and traditional ones. In our case, $O(3^N)$ is computed based on the number of possible combinations of input-output modes. $O(1)$ is computed based on the fact that we only have 2 sets of trainable parameters.

---

### Official Review · Reviewer_5HgX · 2023-10-30

**Soundness:** 2 fair
**Presentation:** 2 fair
**Contribution:** 3 good
**Rating:** 6
**Confidence:** 3

**Summary:**

The paper presents a novel framework, MultiHyperGNN, designed to address challenges in graph transformation, particularly in predicting transitions of graphs across various modes (such as gene expression networks) and generalizing to unseen domains. The primary challenges identified are the high complexity of training on all input-output mode combinations, the difficulty in transforming graphs with differing topologies, and the need to predict graphs in unseen target domains.

MultiHyperGNN introduces a multi-input, multi-output strategy utilizing hypernetworks to generate encoders and decoders, facilitating domain generalization. This approach dramatically reduces space complexity and enables the model to adapt to varying topologies and unseen domains. The framework includes semi-supervised link prediction to enhance output graph completion. Extensive experiments on three real-world datasets demonstrate that MultiHyperGNN outperforms existing models in both prediction accuracy and domain generalization.

--------------------- after reading the authors' comments/rebuttal------------------------------------------
I thank the authors for providing explanations to my questions and answering them satisfactorily. In particular, my concerns regarding the hyperparameter were sufficiently addressed by the experiments the authors conducted by changing the weight on the reconstruction loss for semi-supervised link prediction.

However, I would like to point out two things when the authors were addressing my concerns on the scalability issue. First, it is sufficiently demonstrated that given the largest graph sizes that the model is trained on, scalability doesn't pose any challenges, but my point was to ask if, for even larger graphs, the model is scalable. Second, it is true that the scalability issue is faced by the papers that the authors provided, but the argument "because the field couldn't do it" is not as strong. Nonetheless, I concur that scalability is indeed a difficult topic faced by works about/on graph transformation, and the paper shouldn't be judged unfavorably solely because of it.

The authors argued that meta information is crucial to the domain generalizability and explained in Theorems 1 and 2 to demonstrate the importance that the meta information captures relationships among different domains. The significant improvement over HyperGNN-2 demonstrates the importance of the meta information. However, it's always worth-wondering on the fairness of comparison against approaches to whom meta information (or its variation) is not available. Nonetheless, I do view this as a negative trait of the work as the authors have demonstrated the importance of meta information in their experiment settings.

In summary, I believe this paper demonstrates novelty and valuable info for it to be considered to be accepted. However, when I first reviewed the paper and got to know the strengths and weaknesses of it, I thought the weaknesses are not significant enough to eclipse the strengths. The authors' response addressed some of my concerns. I will maintain my original rating for this paper: marginally above the acceptance threshold

**Strengths:**

+ The multi-input, multi-output hypernetwork-based framework is a novel contribution, particularly in reducing training complexity and addressing domain generalization.
+ Successfully reducing space complexity from exponential to constant during training is a significant advancement, making the model more practical for large-scale applications.
+ The ability to generalize to unseen target domains is a crucial advancement in graph transformation tasks.

**Weaknesses:**

- While space complexity is addressed, the overall computational complexity and potential overfitting risks due to the complex model structure are not discussed in detail.
- The use of hypernetworks might introduce sensitivity to hyperparameters, but this aspect isn’t thoroughly explored.
- The extent to which the model can generalize to radically different unseen domains is not clearly defined.
- It’s unclear how well the model scales to extremely large graphs or how transferable it is across significantly different domains.

**Questions:**

- Can you elaborate on the computational efficiency of MultiHyperGNN, especially when dealing with very large graphs?
- How does the model handle the risk of overfitting, given its complexity?
- Could you discuss any limitations or boundaries in the model's ability to generalize across radically different domains?
- Could you discuss any limitations or boundaries in the model's ability to generalize across radically different domains?

---

> ### Author Response · Authors · 2023-11-19
> **Response to reviewer 5HgX**
>
> We sincerely appreciate the valuable input from the reviewer and giving us the opportunity to further clarify reviewer’s concerns and enhance our paper.
>
> **Comment 1:** While space complexity is addressed, the overall computational complexity and potential overfitting risks due to the complex model structure are not discussed in detail.
>
> **Response 1:**
>
> * We have already discussed the computational complexity in Appendix C:
>
> “Regarding computational complexity, similar to space complexity, ED-GNN, MHM and IN have $O(3^N)$ mode combinations to iterate in each training epoch, where $N$ is the total number of modes in the dataset. The computational complexity of EERN and DRAIN is linear to the number of generators (i.e., $O(2^N)$ generators to handle all combinations of modes) or the number of combinations of time points (i.e., $O(N2^N)$ total number of combinations of time points), which are usually much larger than $N^2$. MultiHyperGNN only needs to train on different input-output mode pairs so that we have only $O(N^2)$ such pairs to iterate in each training epoch, achieving a polynomial computational complexity .”
>
> and we have highlighted this part in red for reviewer’s easier reference.
>
> * Overfitting occurs often when the model is too complex while training data is limited. The total number of input-output mode combinations that our model handles is $O(3^N)$. As a result, we have combinatorially many samples to train just two hypernetworks so that the model is very less likely to overfit.
>
> * We trained our model in the source domain, and evaluated the model in target domain. The source domain and the target domain are exclusive to each other. Our results show a superior performance of our method than comparisons. For instance, for domain generalization, MultiHyperGNN beats DRAIN by 39.09% on average of MSE and 46.23% on average of Pearson correlation coefficient. MultiHyperGNN outperforms EERM by 20.41% on average of MSE and 35.34% on average of Pearson correlation coefficient. This also suggests that our model does not overfit data.
>
> **Comment 2:** The use of hypernetworks might introduce sensitivity to hyperparameters, but this aspect isn’t thoroughly explored.
>
> **Response 2:** We only have one hyperparameter in the model, which is the weight on the reconstruction loss for semi-supervised link prediction. We have conducted experiments by tuning the hyperparameter from 0.01, 0.1, 1 to 10 and the results are as below. The experiments are based on MultiHyperGNN-GIN performed on Genes dataset (same setting as in Table 2):
>
> | weight  | Genes-L |        | Genes-LG |        | Genes-S |        |
> |------|---------|--------|----------|--------|---------|--------|
> |      | MSE     | PCC    | MSE      | PCC    | MSE     | PCC    |
> | 0.01 | 2.1034  | 0.5496 | 2.1455   | 0.5253 | 2.3647  | 0.5050 |
> | 0.1  | 1.8346  | 0.6462 | 1.9990   | 0.6209 | 2.1001  | 0.6165 |
> | 1    | 1.8005  | 0.6600 | 1.9852   | 0.6479 | 1.9031  | 0.6471 |
> | 10   | 1.8100  | 0.6318 | 2.0904   | 0.5866 | 1.9105  | 0.6260 |
>
> The results show a consistent performance when the weight is large, but we do observe the drop of performance when the weight is small (i.e., 0.01), indicating that the semi-supervised learning module to impute output graph is significant for the downstream prediction.
>
> **Comment 3:** The extent to which the model can generalize to radically different unseen domains is not clearly defined.
>
> **Response 3:** We explained this in Theorem 1 and its proof:
>
> * Theorem 1 suggests that the most informative meta information that can maximize the predictibility of $\mathcal G^{(k)}$ will perfectly generalize the model to the domain $k$.
>
> * In Eq. (16) of Appendix, we demonstrate that more informative meta information (i.e., $m$), which leads to a higher probability $p(\\{X_i^{(k)}\\}_{i=1}^n|A^{(k)}, \\{G_i^{(j)}\\} _{i=1}^n, m)$ on the left hand side, results in a reduced generalization error on the right hand side. This theoretically suggests that richer meta information can effectively generalize better to the unseen domain.

---

> ### Author Response · Authors · 2023-11-19
> **Further response to reviewer 5HgX**
>
> **Comment 4:** It’s unclear how well the model scales to extremely large graphs or how transferable it is across significantly different domains.
>
> **Response 4:**
>
> * The scalability to large graphs depends on the scalability of GNN (i.e., the encoder and the decoder), which is not the focus of this paper but can be a future work. This is because our work focuses on hypernetwork that generates encoder and decoder whose numbers of parameters are not correlated to the number of nodes or edges in the graphs. In our experiments, the largest graph size our model is trained on is 2,398, which does not cause any scalability issues. Additionally, many existing works in graph transformation also face the same challenge such as [1] and [2].
>
> * How transferable the model is across significantly different domains depends on how well the meta information captures relationships among domains. If they are not able to capture relationships among different domains, then the domain generalization would still be very hard. We explained this in Theorem 1 and its proof, which suggests that richer meta information can effectively generalize better to the unseen domain.
>
> * We also conducted ablation studies to show the significance of using informative meta information for domain generalization, HyperGNN is a single-to-multiple version of MultiHyperGNN and its variation HyperGNN-2 limits the meta information input to only the mode type. The results show that HyperGNN performs 40.86% better in MSE and 147.59% better in Pearson correlation coefficient than HyperGNN-2, as shown in the domain generalization performance on the Genes dataset (Table 2). This suggests that detailed meta information significantly enhances the capture of relationships among domains and improves domain generalization performance.
>
> [1] Deep multi-attributed graph translation with node-edge co-evolution. ICDM 2019.
>
> [2] Deep graph translation. TNNLS 2022.

---

> > ### Comment · Reviewer_5HgX · 2023-11-21
> >
> > I thank the authors for their efforts in composing the rebuttal. I have updated my official review to reflect my thoughts and my final recommendation for this paper's acceptance/rejection.

---

### Official Review · Reviewer_cBBF · 2023-10-30

**Soundness:** 2 fair
**Presentation:** 2 fair
**Contribution:** 2 fair
**Rating:** 5
**Confidence:** 4

**Summary:**

This article addresses the graph transition problem. And the authors introduce the MultiHyperGNN model, which reduces complexity and enhances model generalization as well. The experimental results on real-world datasets demonstrate the effectiveness and efficiency of the proposed framework. Overall, the paper is a bit hard to understand.

**Strengths:**

1. Solving the graph transformation problem is novel.

2. The experimental results are extensive, covering a wide range of datasets.

**Weaknesses:**

1.	It is unclear about the practical use case of the proposed model, and hard to understand the problem setups. Can you elaborate on the potential real-world applications in which you proposed domain transformation problem can be applied?

2.	It would be beneficial for the authors to explain why traditional methods require O(3^N) to solve the problem. Additionally, is it possible that the traditional method, despite its time-consuming nature, performs the best and can serve as a baseline?

3.	In the experiments on climate dataset (e.g., table 1 and 2), why does HyperGNN use late-night data while MultiGNN uses morning data? Is it a fair comparison?

4.	In the ablation studies, the authors compare with HyperGNN-1 and HyperGNN-2, but do not compare with MultiHyperGNN-1 or MultiHyperGNN-2. I noticed that HyperGNN contains only one decoder, and its structure does not align with the proposed MultiHyperGNN model. Could you please clarify this choice？

5.	In Section 4.3, the definition of meta-information is very strict. It is necessary to discuss how meta-information in the datasets in experiments can satisfy the condition in the definition of meta-information. The authors mentioned that “An ample amount of meta-information will result in reduced generalization error”, but it is unclear of how to measure the “ample amount”, which makes the conclusion too vague.

6.	The notation are overly complex and contain some errors, which hinders readability. For example, in Figure 2, encoder should be represented as f^{(2)}_e. The distinction between \widetilde{A} and \hat{A} in Equations 8 and 9 should be clarified. Moreover, Figure 2 is unclear to me. Whether the decoder's input is a graph or the latent embedding obtained by the encoder, as shown in Equation 11. Why there are two layers of decoder f^{(1)}_d.

**Questions:**

see weakness

---

> ### Author Response · Authors · 2023-11-15
> **Response to reviewer cBBF**
>
> We are grateful for the reviewer's valuable input and the opportunity to correct and enhance our paper. Our clarification to reviewer's questions is as below:
>
> **Comment 1:** Elaborate on the potential real-world applications in which you proposed domain transformation problem can be applied?
>
> **Response 1:**
>
> *  I would like to highlight the use of graph transformation in cross-tissue gene prediction. Variations in gene expression play a significant role in the susceptibility to complex diseases [1], and these expressions are often specific to certain tissues [2]. Our work addresses the challenge of predicting gene expression in less accessible tissues (such as lung or brain) from data obtained from more accessible tissues (like whole blood) [3, 4]. We frame this as a graph transformation task, leveraging the gene-gene interaction networks from source tissue and target tissues. This application of graph transformation in gene prediction is notably emphasized in the Introduction of our paper (i.e., highlighted in red in the first paragraph of Introduction).
>
> * Furthermore, graph transformation has broader applications beyond gene prediction [5]. For example, chemists use it to predict chemical reaction outcomes of the output molecular graph from the input molecular graph [6], computer scientists apply it to distinguish between normal and compromised user authentication graphs [7], and urban planners utilize it for forecasting on traffic networks at different time points [8]. These diverse applications, cited in our submission ([5, 6, 7, 8]), demonstrate the versatility of graph transformation. Additionally, those tasks are of one-to-one graph transformation but not multiple-to-multiple because of time and space complexity, but we push it forward to multiple-to-multiple so that it can leverage more of the mode-specific information.
>
> * To evaluate our model, we have incorporated datasets relevant to these tasks: the Genes dataset for cross-tissue gene prediction, the Climate dataset for temporal weather prediction, and the Traffic dataset for traffic flow prediction. The results, as shown in Tables 1 and 2, validate our model's efficacy across these varied applications. Notably, in Genes dataset, our model exhibits an average MSE 4.24% lower than the second-best model. Our model also has the Pearson correlation coefficient 4.74% higher by average than the second best model. In traffic forecasting, the proposed model surpasses the second-best model by 7.62% and 1.97% on average in MSE and Pearson correlation coefficient, respectively, for domain generalization.
>
> [1] Mapping complex disease traits with global gene expression.Nature Reviews Genetics 2009. [2] Understanding tissue-specific gene regulation. Cell reports 2021.
>
> [3] Predicting tissue-specific gene expression from whole blood transcriptome. Science Advances 2021.
>
> [4] Prediction of the gene expression in normal lung tissue by the gene expression in blood. BMC medical genomics 2015.
>
> [5] Graph Neural Networks: Graph Transformation. GNNs: Foundations, Frontiers, and Applications. 2021.
>
> [6] Deep multi-attributed graph translation with node-edge co-evolution. ICDM 2019.
>
> [7] Deep graph translation. TNNLS 2022.
>
> [8] Spatio-Temporal Graph Convolutional Networks: A Deep Learning Framework for Traffic Forecasting. IJCNN 2018.

---

> > ### Author Response · Authors · 2023-11-16
> > **Further response to reviewer cBBF**
> >
> > **Comment 6:** The notation are overly complex and contain some errors, which hinders readability. For example, in Figure 2, encoder should be represented as $f^{(2)}_e$. The distinction between $\widetilde{A}$ and $\hat{A}$ in Equations 8 and 9 should be clarified. Moreover, Figure 2 is unclear to me. Whether the decoder's input is a graph or the latent embedding obtained by the encoder, as shown in Equation 11. Why there are two layers of decoder $f^{(1)}_d$.
> >
> > **Response 6:**
> >
> > * For further clarification on the notation, we have added a notation table in Appendix Table 3 as a reference for our readers.
> >
> > * We have corrected the typo in Figure 1, changing $f_e^{(1)}$ to $f_e^{(2)}$.
> >
> > * The paper also addresses the challenge of predicting for graphs with different topologies in the source and target modes. We employ semi-supervised link prediction to infer edges between disjoint nodes. The final topology, $\hat{A}$, learned through semi-supervised link prediction and supervised by $\tilde{A}$ (the output mode's adjacency matrix), is detailed in Eq. (8) and (9), now highlighted in red in the revised paper.
> >
> > * Yes, the decoder's input is a graph or the latent embedding obtained by the encoder. The decoder's role is two-fold: it first joins the potentially different topologies of source and output modes through semi-supervised link prediction, and then predicts values for all nodes. In this process, the first module of $f_d^{(1)}$ encodes the output mode's graph using node embeddings from the encoder, focusing on nodes present in the source mode. Subsequently, the second module of it performs semi-supervised link prediction for nodes outside the source mode and predicts final values for all nodes in the output mode.

---

> ### Author Response · Authors · 2023-11-15
> **Further response to reviewer cBBF**
>
> **Comment 2:** Explain why traditional methods require $O(3^N)$ to solve the problem. Additionally, is it possible that the traditional method, despite its time-consuming nature, performs the best and can serve as a baseline?
>
> **Response 2:** We have elaborated on the complexity of our method and traditional methods in Section 4.1 and Appendix C of our paper, respectively, as highlighted in red.
>
> * Traditional approaches require a complexity of $O(3^N)$ to address the problem, given that each of the N modes can serve as either a source, a target, or neither, leading to $3^N$ possible combinations. While these methods could be effective, they incur substantial costs in terms of parameter space and training time complexity.
>
> * We also compared our method with those that leverage hypernetworks (i.e., DRAIN) or adversarial traing (i.e., EERM) for domain generalization. MultiHyperGNNs shows consistently better performance compared with other models (i.e., Table 1 and Table 2). For instance, for domain generalization, MultiHyperGNN beats DRAIN by 39.09% on average of MSE and 46.23% on average of Pearson correlation coefficient. MultiHyperGNN outperforms EERM by 20.41% on average of MSE and 35.34% on average of Pearson correlation coefficient.
>
> * For optimal performance, these models necessitate training $O(3^N)$ encoder-decoder pairs to handle all mode combinations. Considering large datasets like the GTEx dataset, which includes over 45 tissue types (i.e., modes), this approach would require training an impractical number of combinations ($3^{45} \approx 3 \times 10^{21}$). Additionally, the lack of aligned training data for some mode combinations further complicates this approach, often rendering it infeasible.
>
> * In contrast, our method processes arbitrary combinations of input and output modes efficiently. We leverage hypernetworks to dynamically generate the necessary encoders and decoders, addressing the challenge of handling the exponentially growing number of input-output mode combinations. This approach significantly reduces the computational burden and makes it feasible to work with large and complex datasets.
>
> **Comment 3:** In the experiments on climate dataset (e.g., table 1 and 2), why does HyperGNN use late-night data while MultiGNN uses morning data?
>
> **Response 3:** We are grateful for the reviewer's valuable input and the opportunity to correct and enhance our paper. I would like to clarify that HyperGNN, along with its variants EERM and DRAIN, are trained from late morning data only. This detail has been corrected and highlighted in the revised version of our paper.

---

> ### Author Response · Authors · 2023-11-15
> **Further response to reviewer cBBF**
>
> **Comment 4:** In the ablation studies, the authors compare with HyperGNN-1 and HyperGNN-2, but do not compare with MultiHyperGNN-1 or MultiHyperGNN-2. I noticed that HyperGNN contains only one decoder, and its structure does not align with the proposed MultiHyperGNN model. Could you please clarify this choice？
>
> **Response 4:**: We are grateful for the reviewer's valuable feedback, which has provided us an opportunity to clarify aspects of our study and enhance our paper.
>
> * In the ablation study, HyperGNN serves as a simplified version of MultiHyperGNN, designed to predict multiple output modes from a single input mode. Therefore it contains only one encoder.
>
> * To assess the importance of generating the final prediction layer via hypernetworks, we introduced HyperGNN-1, which learns this layer instead of producing it through hypernetworks. For a fair comparison, we evaluated HyperGNN-1 against HyperGNN rather than MultiHyperGNN. This comparison more directly tests the model's ability to learn all mode combinations, which is a challenging task. Our results, particularly the domain generalization performance on the Genes dataset (shown in Table 2), indicate that HyperGNN significantly outperforms HyperGNN-1, with improvements of 42.74% in MSE and 148.55% in Pearson correlation coefficient on average.
>
> * Further, to test the hypothesis that more informative meta information yields better performance, we developed HyperGNN-2, which limits the meta information to only the mode type. Comparing HyperGNN-2 with HyperGNN, we again observed superior performance of HyperGNN in all settings. Specifically, HyperGNN shows 40.86% better MSE and 147.59% better Pearson correlation coefficient on average than HyperGNN-2, as evidenced by the domain generalization performance on the Genes dataset in Table 2.
>
> * In summary, both HyperGNN-1 and HyperGNN-2 were introduced to validate the necessity of generating prediction layers from hypernetworks and utilizing informative meta information. They were compared with HyperGNN, not MultiHyperGNN. This discussion is highlighted in the second paragraph of Section 5.4.
>
> * Additionally, we are conducting further experiments with MultiHyperGNN-1 and MultiHyperGNN-2 compared with MultiHyperGNN, which learns the prediction layer instead of producing it through hypernetworks and limits the meta information to only the mode type, respectively. The experimemts are based on MultiHyperGNN-GIN on Genes dataset regarding domain generalization (same experimental settings as Table 2). Below are the resutls:
>
> | Model               | Genes-L |        | Genes-LG |        | Genes-S |        |
> |-------------------|---------|--------|----------|--------|---------|--------|
> |                   | MSE     | PCC    | MSE      | PCC    | MSE     | PCC    |
> | MultiHyperGNN | **1.8005**  | **0.6600** | **1.9852**   | **0.6479** | **1.9031**  | **0.6471** |
> | MultiHyperGNN-1   | 3.5236  | 0.2795 | 2.9149   | 0.3997 | 2.9512  | 0.4005 |
> | MultiHyperGNN-2   | 3.0071  | 0.4114 | 3.0294   | 0.3316 | 3.1411  | 0.3130 |
>
> Specifically, MultiHyperGNN outperforms MultiHyperGNN-1 by 39.70% regarding MSE and 80.98% regarding Pearson correlation coefficient on average, respectively. This indicates the importance of generating the final prediction layer via hypernetworks as training a single prediction layer may not well handel exponential number of mode combinations.
>
> Similarly, MultiHyperGNN outperforms MultiHyperGNN-2 by 38.31% regarding MSE and 85.04% regarding Pearson correlation coefficient on average, respectively. This indicates that more informative meta information could help with better model performance on domain generalization.

---

> ### Author Response · Authors · 2023-11-19
> **Further response to reviewer cBBF**
>
> **Comment 5:** In Section 4.3, the definition of meta-information is very strict. It is necessary to discuss how meta-information in the datasets in experiments can satisfy the condition in the definition of meta-information. The authors mentioned that “An ample amount of meta-information will result in reduced generalization error”, but it is unclear of how to measure the “ample amount”, which makes the conclusion too vague.
>
> **Response 5:** We thank the reviewer for their insightful feedback and the opportunity to further clarify and enhance our paper.
>
> * Ample meta information means the most informative meta data for guiding the generating of $\mathcal G^{(k)}$. It guides us how to select meta information, namely we select those that can maximize the predictibility of $\mathcal G^{(k)}$ and don't need to select those that cannot help the predictibility of $\mathcal G^{(k)}$.
>
> * In practice, this theorem can guide us in several ways. First, we can try to solve this optimization. In some situation, we have prior knowledge that some meta information must be very important and some should be irrelevant so can guide our selection. In Eq. (16) of Appendix, we demonstrate that more informative meta information (i.e., $m$), which leads to a higher probability $p(\\{X_i^{(k)}\\}_{i=1}^n|A^{(k)}, \\{G_i^{(j)}\\} _{i=1}^n, m)$ on the left hand side, results in a reduced generalization error on the right hand side. This theoretically suggests that richer meta information can effectively lower generalization error.
>
> * For example, in cross-tissue gene prediction, HyperGNN-2, using only tissue type as meta information, underperforms compared to HyperGNN, which employs richer meta information including tissue type, location, structure, and cell types (as shown in Tables 1 and 2). The optimal meta information selection can be determined through general model selection techniques, such as cross-validation, or domain expertise.

---

> > ### Comment · Reviewer_cBBF · 2023-11-23
> > **Thanks for your response**
> >
> > Thank for providing detailed responses. My previous confusion has been partially resolved. I suggest the authors to further polish the draft and make it more readable. I will raise my score from 3 to 5.

---

> > > ### Author Response · Authors · 2023-11-23
> > > **Further response to reviewer cBBF**
> > >
> > > We appreciate the feedback from the reviewer that helps us further improve our paper. Please let us know if there are still any concerns and we are very happer to clarify. To make the paper more readable, we have added a notation table in Appendix Table 3 as a reference for our readers. We believe this could large improve the readability of the paper.

---

### Official Review · Reviewer_tWqJ · 2023-10-31

**Soundness:** 2 fair
**Presentation:** 3 good
**Contribution:** 3 good
**Rating:** 6
**Confidence:** 3

**Summary:**

The paper introduces a novel approach to graph transformation, specifically focusing on domain generalization. Traditional graph transformation techniques often assume that testing and training data have the same distribution, which is not always the case. The authors address challenges like space complexity, differences in graph topologies, and generalizing to unseen domains. They propose a multi-input, multi-output, hypernetwork-based graph neural network (MultiHyperGNN) that uses an encoder and a decoder to handle both input and output modes. The model also incorporates semi-supervised link prediction to enhance the transformation task. Experiments demonstrate that MultiHyperGNN outperforms competing models in prediction and domain generalization tasks.

**Strengths:**

- Originality: The paper addresses the under-explored area of domain generalization in graph transformation. And introduce MultiHyperGNN, a multi-input, multi-output hypernetwork-based GNN.
- Quality: The proposed model effectively tackles the challenges of space complexity, varying graph topologies, and generalization to unseen domains. And comprehensive experiments validate the effectiveness of MultiHyperGNN.
- Clarity: The paper is well-structured, with a clear presentation of the problem, challenges, and the proposed solution.
Significance:

The work has potential implications for various applications where domain generalization is crucial, enhancing the robustness and applicability of graph transformation models.

**Weaknesses:**

1.  The introduction part seems to be incomplete. In the third paragraph, the three challenges of the problem are emphasized. The fourth paragraph should focus on writing the core idea of ​​solving these challenges in this work.
However, the authors only wrote about their implementation, so I did not understand the innovative ideas of this work from a high-level perspective.
2.  The paper does not adding ablation studies to understand the contribution of each component of MultiHyperGNN.
3.  While the challenges are listed, a more in-depth discussion on how each challenge is specifically addressed by MultiHyperGNN would enhance understanding.

**Questions:**

1.  How do the proposed hypernetworks differ from existing ones in terms of architecture and functionality?
2.  Consider adding ablation studies to understand the contribution of each component of MultiHyperGNN.

---

> ### Author Response · Authors · 2023-11-17
> **Response to reviewer tWqJ**
>
> Thank you for your valuable feedback and giving us the opportunity to clarify your concerns.
>
> **Comment 1:** The introduction part seems to be incomplete. In the third paragraph, the three challenges of the problem are emphasized. The fourth paragraph should focus on writing the core idea of ​​solving these challenges in this work. However, the authors only wrote about their implementation, so I did not understand the innovative ideas of this work from a high-level perspective.
>
> **Response 1:** We have detailed our strategies in the fourth paragraph of the revised paper, now highlighted in red for easier reference. Here are the specifics:
>
> * **Challenge 1 - High Complexity in the Training Process:** “We introduced a novel framework leveraging a multi-input, multi-output training strategy, significantly reducing the space complexity regarding trainable parameters from exponential to constant during training.”
>
> * **Challenge 2 - Graph Transformation Between Topologically Different Modes:** “We develop an encoder and a decoder for graph transformation between topologically different input and output modes, respectively. Additionally, our model conducts semi-supervised link prediction to complete the output graph, facilitating generalization to all nodes in the output mode.”
>
> * **Challenge 3 - Learning Graph Transformation Involving Unseen Domains and Lack of Training Data:** “We design two novel hypernetworks that produce the encoder and the decoder for domain generalization. Mode-specific meta information serves as the input to guide the hypernetwork to produce the corresponding encoder or decoder, and generalize to unseen target domains.”
>
> **Comment 2:** The paper does not add ablation studies to understand the contribution of each component of MultiHyperGNN.
>
> **Response 2:** Thank you for your interest in our ablation studies. We conducted these studies, HyperGNN, HyperGNN-1 and HyperGNN-2 to rigorously test various aspects of our model. Below are the details of our findings:
>
> * **HyperGNN**. To evaluate whether more input domains will result in better predictive performance. HyperGNN is a simpler version of MultiHyperGNN that predicts multiple output modes from a single input mode. In gene prediction tasks, MultiHyperGNN surpassed HyperGNN by an average of 4.24% in MSE and 4.74% in Pearson correlation coefficient. This demonstrates the advantages of using multiple input modes for domain generation in graph transformation.
>
> * **HyperGNN-1**. To evaluate if a single MLP prediction layer is able to predict all output modes rather than being learned from the decoder hypernetwork, unlike HyperGNN, HyperGNN-1 does not generate MLP layers but only GNN layers via a hypernetwork. The domain generalization performance on the Genes dataset (Table 2) shows that HyperGNN significantly outperforms HyperGNN-1, with a 42.74% improvement in MSE and 148.55% in Pearson correlation coefficient. This indicates the benefit of using mode-specific prediction layers.
>
> * **HyperGNN-2**. To determine the importance of using informative meta information for domain generalization, HyperGNN-2 limits the meta information input to only the mode type. The results show that HyperGNN performs 40.86% better in MSE and 147.59% better in Pearson correlation coefficient than HyperGNN-2, as shown in the domain generalization performance on the Genes dataset (Table 2). This suggests that detailed meta information significantly enhances the capture of relationships among domains and improves domain generalization performance.

---

> ### Author Response · Authors · 2023-11-17
> **Further response to reviewer tWqJ**
>
> **Comment 3:** While the challenges are listed, a more in-depth discussion on how each challenge is specifically addressed by MultiHyperGNN would enhance understanding.
>
> **Response 3:**:
>
> * Challenge 1 - High Complexity in the Training Process:
>
>     - In our contribution list under paragraph four, we claimed our strategy to address the heavy complexity of traditional methods: “We introduced a novel framework leveraging a multi-input, multi-output training strategy, significantly reducing the space complexity regarding trainable parameters from exponential to constant during training.”
>
>     - In the last paragraph of Section 4.1, we have explained how MultiHyperGNN could reduce space complexity. This part has been highlighted in red in the revised paper for the easier reference for the reviewer.
>
>     - We conducted complexity analysis in Appendix C, which is highlighted in red for easier reference. The analysis shows that our method is able to achieve a constant training space complexity, which is the same as DRAIN, but superior than other baselines. Additionally, our method is able to achieve a polynomial (i.e., $O(N^2)$) training computational complexity, which is better than other baselines as well.
>
> * Challenge 2 - Graph Transformation Between Topologically Different Modes:
>
>     - In our contribution list under paragraph four, we have explained our strategy to handle topologically different input and output modes: “We develop an encoder and a decoder for graph transformation between topologically different input and output modes, respectively. Additionally, our model conducts semi-supervised link prediction to complete the output graph, facilitating generalization to all nodes in the output mode.”
>
>     - In the Section 4.2, we introduced the detailed framework of GNN-based encoder and decoder to handle different input and output topologies. We also introduced how we perform semi-supervised link prediction to predict nodes that are not overlapped by the input and output graphs.
>
>     - We also conduced another ablation study by replacing GNN in the encoder and the decoder with MLP, i.e., MultiHyperGNN-MLP, which cannot well learn the graph topologies of input and output modes. The results in Table 1 and Table 2 show that MultiHyperGNN-MLP has a much worse predictive and domain generalization performation compared with MultiHyperGNN.
>
> * Challenge 3 - Learning Graph Transformation Involving Unseen Domains and Lack of Training Data:
>
>     - In our contribution list under paragraph four, we claimed our strategy for domain generalization by leveraging relationships among domains via meta information: “We design two novel hypernetworks that produce the encoder and the decoder for domain generalization. Mode-specific meta information serves as the input to guide the hypernetwork to produce the corresponding encoder or decoder, and generalize to unseen target domains.”
>
>     - In Section 4.3, we introduced detailed design of hypernetworks that take domain-specific meta information as the input to guide the domain generalization. Particularly, to guide the selection of meta information, we introduced a theory (i.e., Theorem 1) and its proof to indicate that informative meta data for guiding the generating of the output mode could help reduce the generalization error.
>
>     - We also conducted ablation study to address the significance of using informative meta information for domain generalization. HyperGNN-2 limits the meta information input to only the mode type. The results show that HyperGNN performs 40.86% better in MSE and 147.59% better in Pearson correlation coefficient than HyperGNN-2, as shown in the domain generalization performance on the Genes dataset (Table 2). This suggests that detailed meta information significantly enhances the capture of relationships among domains and improves domain generalization performance.

---

### Meta-Review · Area_Chair_QEFs · 2023-12-06

**Metareview:**

The paper presents a new approach to graph transformation, specifically focusing on domain generalization. Specifically, the authors identified three key challenges and proposed a multi-input, multi-output hypernetwork-based graph neural networks (MultiHyperGNN). Reviewers agreed that this paper investigates a relatively under-explored area of domain generalization, and the proposed method is well motivated. Experimental results show the advantages of the proposed framework on some benchmark datasets.

While the problem looks interesting and under-explored, the solution looks a bit engineered. Reviewers raised many concerns regarding technical details (e..g, optimization), experiments (e.g., hyperparameter settings, ablation studies), paper presentation (e.g., definition of meta-information, math notations), etc. Many of these concerns have been addressed by the detailed responses from authors. However, during the discussion stage, reviewer still concerned about the scalability, mathematical notations, empricial comparisons, etc. Hopefully the authors can incorporate the suggestions from reviewers into the future version of this paper.

**Justification For Why Not Higher Score:**

This paper studies an important problem and presents an interesting approach. However, the current version still has some limitations, such as paper presentation and experiments.

**Justification For Why Not Lower Score:**

N/A

---

### Decision · Program_Chairs · 2024-01-16

Reject